# Coarse-to-fine Q-Network with Action Sequence for Data-Efficient Reinforcement Learning

**Younggyo Seo**
UC Berkeley
mail@younggyo.me

**Pieter Abbeel**
UC Berkeley
pabbeel@cs.berkeley.edu

## Abstract

Predicting a sequence of actions has been crucial in the success of recent behavior cloning algorithms in robotics. Can similar ideas improve reinforcement learning (RL)? We answer affirmatively by observing that incorporating action sequences when predicting ground-truth return-to-go leads to lower validation loss. Motivated by this, we introduce Coarse-to-fine Q-Network with Action Sequence (CQN-**AS**), a novel value-based RL algorithm that learns a critic network that outputs Q-values over a sequence of actions, i.e., explicitly training the value function to learn the consequence of executing action sequences. Our experiments show that CQN-**AS** outperforms several baselines on a variety of sparse-reward humanoid control and tabletop manipulation tasks from BiGym and RLBench. Code is available at:
https://younggyo.me/cqn-as/

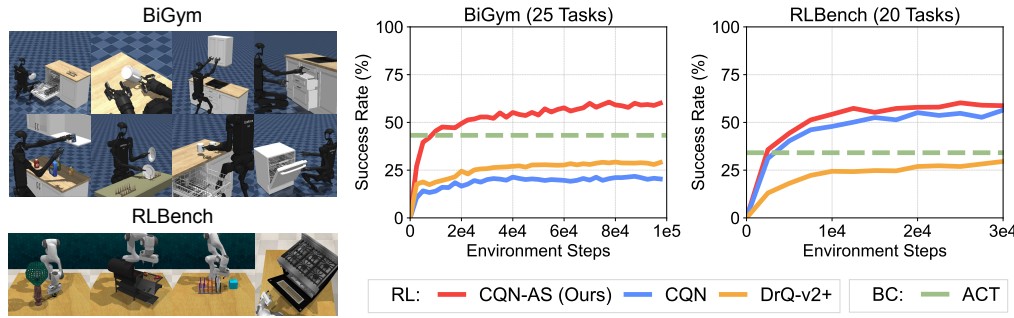

Figure 1: **Summary of results.** Coarse-to-fine Q-Network with **A**ction **S**equence (CQN-**AS**) learns a critic network with action sequence. CQN-**AS** outperforms various RL and BC baselines such as CQN (Seo et al., 2024), DrQ-v2+ (Yarats et al., 2022), and ACT (Zhao et al., 2023) on 45 robotic tasks from BiGym (Chernyadev et al., 2024) and RLBench (James et al., 2020).

## 1 Introduction

Predicting action sequences from expert trajectories is a key idea in recent successful behavior cloning (BC; Pomerleau 1988) approaches in robotics. This has enabled policies to effectively approximate the noisy, multi-modal distribution of expert demonstrations (Zhao et al., 2023; Chi et al., 2023). Can this idea similarly be useful for reinforcement learning (RL)?

Our initial finding is affirmative: we make an intriguing observation that using action sequences can enhance value learning. Specifically, with humanoid demonstrations from BiGym (Chernyadev et al., 2024), we train regression models that predict the ground-truth return-to-go, i.e., the sum of discounted future rewards from the timestep $t$, given the current observation and action. In Figure 2a, we find that using an action sequence $\mathbf{a}_{t:t+K} = \{\mathbf{a}_t, ..., \mathbf{a}_{t+K-1}\}$ as input results in lower validation

39th Conference on Neural Information Processing Systems (NeurIPS 2025).

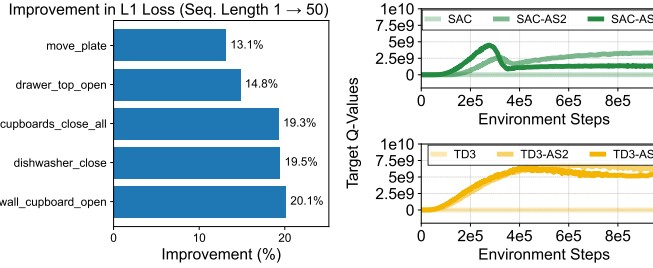

| | | TD3 | | CQN | |
|---|---|---|---|---|---|
| No-op added | | Return | Value | Return | Value |
| 0 | | 140.7 | 16.25 | 236.6 | 18.8 |
| 54 | | 7.18 | -17.45 | 219.5 | 20.3 |
| 144 | | 0.56 | 1E8 | 185.4 | 16.4 |
| 294 | | 0.27 | 4E8 | 202.9 | 18.7 |

(a) Return-to-go prediction     (b) Value overestimation on `stand`     (c) Effect of no-op actions

Figure 2: **Analyses.** (a) We measure the improvement in the validation L1 loss of the return-to-go prediction model with different action sequence lengths. We find that using action sequence of length 50 results in the lower loss than using single-step action. (b) We find that SAC and TD3 with action sequences suffer from severe value overestimation in `stand` task from HumanoidBench, which leads to random near-zero performance. (c) Actor-critic algorithms like TD3 become vulnerable to value overestimation when redundant no-op actions are added to the action space. In contrast, a critic-only algorithm that uses discrete actions, CQN, is robust with high-dimensional action spaces.

losses than using a single-step action $\mathbf{a}_t$. We hypothesize this is because action sequences, which can correspond to behavioral primitives such as *going straight*, make it easier for the model to learn the long-term outcomes compared to evaluating the effect of individual single-step actions (see Appendix A for additional analysis based on a 2D Point-mass environment).

Building on this observation, we train actor-critic algorithms (Haarnoja et al., 2018; Fujimoto et al., 2018) with action sequence on `stand` task from HumanoidBench (Sferrazza et al., 2024). Specifically, we train the actor to output action sequence and the critic to take action sequence as inputs instead of single-step actions. However, we find these algorithms with action sequences suffer from severe value overestimation (see Figure 2b) and completely fail to solve the task. This is because a wider action space makes the critic more vulnerable to function approximation error (Fujimoto et al., 2018) and the actor excessively maximizes value functions by exploiting this estimation error. To further support this, in Figure 2c, we report additional toy experiments where we introduce redundant no-op actions for training TD3 agents on `Cheetah Run` task (Tassa et al., 2020). Here, we find that actor-critic algorithms are indeed vulnerable to value overestimation with high-dimensional action spaces.

This result motivates us to design our RL algorithm with action sequence upon a recent critic-only algorithm, i.e., Coarse-to-fine Q-Network (CQN; Seo et al. 2024), which solves continuous control tasks with discrete actions. Because there is no separate actor that may exploit value functions, i.e., CQN simply selects discrete actions with the highest Q-values, we find that training with action sequences is stable and thus avoids value overestimation problem (see Figure 2c). In particular, we introduce Coarse-to-fine Q-Network with **A**ction **S**equence (CQN-**AS**), which learns a critic network that outputs Q-values over a sequence of actions (see Figure 3). By training the critic network to explicitly learn the consequence of taking a sequence of current and future actions, CQN-**AS** enables the RL agents to effectively learn useful value functions on challenging robotic tasks.

Our experiments show that CQN-**AS** improves the performance of CQN on sparse-reward humanoid control tasks from BiGym benchmark (Chernyadev et al., 2024) that provides human-collected demonstrations and sparse-reward tabletop manipulation tasks from RLBench (James et al., 2020) that provide demonstrations generated via motion-planning. Considering that CQN-**AS** is a critic-only algorithm that selects actions with the highest Q-value without a separate actor network, these results highlight the benefit of using action sequences in value learning.

Our contributions can be summarized as below:

- We make an observation that shows using action sequences can be useful for RL by enhancing value learning. We also show that standard actor-critic algorithms (Haarnoja et al., 2018; Fujimoto et al., 2018) suffer from value overestimation when trained with action sequences.
- We introduce Coarse-to-fine Q-Network with Action Sequence (CQN-**AS**) that trains a critic network to output Q-values over action sequences. This critic-only algorithm successfully avoids value overestimation problem and enhances the base CQN algorithm.

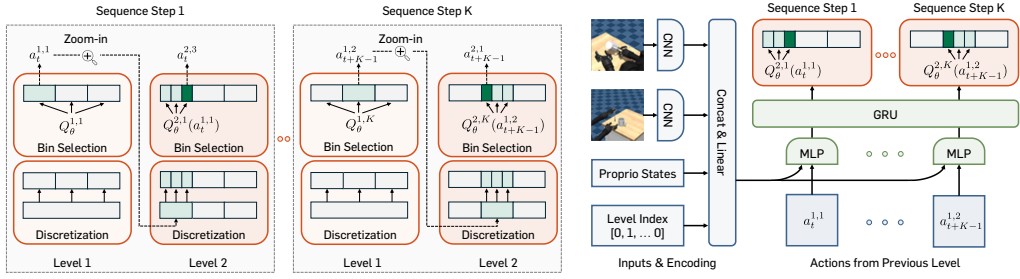

(a) Coarse-to-fine inference procedure      (b) Architecture

Figure 3: **Coarse-to-Fine Q-Network with Action Sequence.** CQN-**AS** extends Coarse-to-Fine Q-Network (CQN; Seo et al. 2024), a critic-only RL algorithm for continuous control using discretized actions. (a) CQN progressively zooms into the action space by discretizing it into $B$ bins and finding the bin with the highest Q-value to further discretize at the next level. Last level's action sequence is used for controlling robots. CQN-**AS** generalizes this to action sequences by computing all $K$ actions in parallel. (b) We train a critic to predict Q-values over entire action sequences by extracting per-step features and aggregating them with a recurrent network before projection to Q-values.

- In a demo-driven RL setup that initializes training with expert demonstrations, we show that CQN-**AS** surpasses the performance of ACT (Zhao et al., 2023) – a BC algorithm that trains a Transformer (Vaswani et al., 2017) to predict action sequences.

## 2   Preliminaries

We formulate a robotic control problem as a partially observable Markov decision process (Kaelbling et al., 1998; Sutton & Barto, 2018). At time step $t$, an RL agent encounters an observation $\mathbf{o}_t$, executes an action $a_t$, receives a reward $r_{t+1}$, and encounters a new observation $\mathbf{o}_{t+1}$ from the environment. We aim to train a policy $\pi$ that maximizes the expected sum of rewards through RL while using as few online samples as possible, optionally with access to a modest amount of expert demonstrations.

**Inputs and encoding**   Given visual observations $\mathbf{o}_t^v = \{\mathbf{o}_t^{v_1}, ..., \mathbf{o}_t^{v_M}\}$ from $M$ cameras, we encode each $\mathbf{o}_t^{v_i}$ using convolutional neural networks (CNN) into $\mathbf{h}_t^{v_i}$. We then process them through a series of linear layers to fuse them into $\mathbf{h}_t^v$. If low-dimensional observations $\mathbf{o}_t^{\texttt{low}}$ are available along with visual observations, we process them through a series of linear layers to obtain $\mathbf{h}_t^{\texttt{low}}$. We then use concatenated features $\mathbf{h}_t = [\mathbf{h}_t^v, \mathbf{h}_t^{\texttt{low}}]$ as inputs to the critic network. In domains without vision sensors, we simply use $\mathbf{o}_t^{\texttt{low}}$ as $\mathbf{h}_t$ without encoding the low-dimensional observations.

**Coarse-to-fine Q-Network**   Coarse-to-fine Q-Network (CQN; Seo et al. 2024) is a critic-only RL algorithm that solves continuous control tasks with discrete actions. CQN trains an RL agent to learn to select coarse discrete actions in shallower levels with larger bin sizes, and then refine their choices by selecting finer-grained actions in deeper levels with smaller bin sizes. Specifically, CQN iterates the procedures of (i) discretizing the continuous action space into multiple bins and (ii) selecting the bin with the highest Q-value to further discretize. This reformulates the continuous control problem as a multi-level discrete control problem, allowing for the use of ideas from sample-efficient discrete RL algorithms (Mnih et al., 2015; Silver et al., 2017) for continuous control.

Formally, let $a_t^l$ be an action at level $l$ with $a_t^0$ being the zero vector.[1] We then define the coarse-to-fine critic to consist of multiple Q-networks which compute Q-values for actions at each level $a_t^l$, given the features $\mathbf{h}_t$ and actions from the previous level $a_t^{l-1}$, as follows:

$$Q_\theta^l(\mathbf{h}_t, a_t^{l-1}) = \left[Q_\theta^l(\mathbf{h}_t, a_t^l = a_t^{l,b}, a_t^{l-1})\right]_{b=1}^B \in \mathbb{R}^B \tag{1}$$

---

[1]For simplicity, we describe CQN and CQN-**AS** with a single-dimensional action in the main section. See Appendix C for full description with $N$-dimensional actions.

where $a_t^{l,b}$ denotes an action for each bin $b$ and $B$ is the number of bins for each level. We note that CQN uses scalar values representing the center of each bin for previous level's action $a_t^{l-1}$, enabling the network to locate itself without access to all previous levels' actions. We optimize each Q-network at level $l$ with the following objective:

$$\mathcal{L}^l = \left(Q_\theta^l(\mathbf{h}_t, a_t^l, a_t^{l-1}) - r_{t+1} - \gamma \max_{a'} Q_{\bar{\theta}}^l(\mathbf{h}_{t+1}, a', \pi^{l-1}(\mathbf{h}_{t+1}))\right),$$

where $\bar{\theta}$ are delayed parameters for a target network (Polyak & Juditsky, 1992) and $\pi^l$ is a policy that outputs the action $a_t^l$ at each level $l$ via the inference steps with our critic, *i.e.*, $\pi^l(\mathbf{h}_t) = a_t^l$. Specifically, to output actions at time step $t$, CQN first initializes constants $a_t^{\texttt{low}}$ and $a_t^{\texttt{high}}$ with $-1$ and $1$. Then the following steps are repeated for $l \in \{1, ..., L\}$:

- Step 1 (Discretization): Discretize an interval $[a_t^{\texttt{low}}, a_t^{\texttt{high}}]$ into $B$ uniform intervals, and each of these intervals become an action space for $Q_\theta^l$.

- Step 2 (Bin selection): Find a bin with the highest Q-value and set $a_t^l$ to the centroid of the bin.

- Step 3 (Zoom-in): Set $a_t^{\texttt{low}}$ and $a_t^{\texttt{high}}$ to the minimum and maximum of the selected bin, which intuitively can be seen as zooming-into each bin.

We then use the last level's action $a_t^L$ as the action at time step $t$. For more details, including the inference procedure for computing Q-values, we refer readers to Appendix C.

## 3 Method

We present Coarse-to-fine Q-Network with **A**ction **S**equence (CQN-**AS**), a value-based RL algorithm that learns a critic network that outputs Q-values for *a sequence of actions* $a_{t:t+K} = \{a_t, ..., a_{t+K-1}\}$ for a given observation $\mathbf{o}_t$. Our main motivation comes from one of the key ideas in recent BC approaches: predicting *action sequences*, which helps resolve ambiguity when approximating noisy distributions of expert demonstrations (Zhao et al., 2023; Chi et al., 2023). Similarly, by explicitly learning Q-values of a sequence of actions from the given state, our approach mitigates the challenge of learning Q-values with noisy trajectories. We provide the overview of CQN-**AS** in Figure 3.

### 3.1 Coarse-to-fine Critic with Action Sequence

**Objective** Let $a_{t:t+K}^l = \{a_t^l, ..., a_{t+K-1}^l\}$ be an action sequence at level $l$ and $a_{t:t+K}^0$ be a zero vector. We design our coarse-to-fine critic network to consist of multiple Q-networks that compute Q-values for each action at sequence step $k \in \{1, ..., K\}$ and level $l \in \{1, ..., L\}$:

$$Q_\theta^{l,k}(\mathbf{h}_t, a_{t:t+K}^{l-1}) = \left[Q_\theta^{l,k}(\mathbf{h}_t, a_{t+k-1}^{l,b}, a_{t:t+K}^{l-1})\right]_{b=1}^{B} \in \mathbb{R}^B$$

where $a_{t+k-1}^{l,b}$ denotes an action for each bin $b$ at step $k$. We optimize our critic network with the following objective:

$$\sum_k \sum_l \left(Q_\theta^{l,k}(\mathbf{h}_t, a_{t+k-1}^l, a_{t:t+K}^{l-1}) - \sum_{i=1}^N r_{t+i} - \gamma \max_{a'} Q_{\bar{\theta}}^{l,k}(\mathbf{h}_{t+1}, a', \pi_K^{l-1}(\mathbf{h}_{t+1}))\right)^2, \quad (2)$$

where $N$ is a hyperparameter for $N$-step return and $\pi_K^l$ is an action sequence policy that outputs the action sequence $\mathbf{a}_{t:t+K}^l$ by following the similar inference procedure as in Section 2 (see Figure 3a). In practice, we compute Q-values for all sequence step $k \in \{1, ..., K\}$ in parallel, which is possible as Q-values for future actions depend only on features $\mathbf{h}_t$ but not on previous actions.

**Remarks on objective with $N$-step return** We note that any $N$-step return can be used in Equation 2 because the network can learn the long-term value of outputting action $a_{t+k}$ from bootstrapping. There is a trade-off: if one considers a short $N$-step return, it can cause a challenge as the setup becomes a delayed reward setup; but training with higher $N$-step return may introduce variance (Sutton & Barto, 2018). In our considered setups, we empirically find that using common values $N \in \{1, 4\}$ works the best. We provide empirical analysis on the effect of $N$ in Figure 8c.

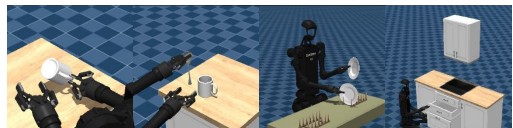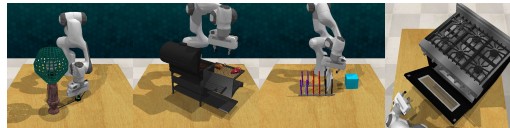

Figure 4: **Examples of robotic tasks.** We study CQN-**AS** on 25 humanoid control tasks from BiGym (Chernyadev et al., 2024) and 20 tabletop manipulation tasks from RLBench (James et al., 2020).

**Architecture**    Our critic network initially extracts features for each sequence step $k$ and aggregates features from multiple steps with a recurrent network (see Figure 3b). This architecture is helpful in cases where a single-step action is already high-dimensional so that concatenating them make inputs too high-dimensional. Specifically, let $\mathbf{e}_k$ denote an one-hot encoding for $k$. At each level $l$, we construct features for each sequence step $k$ as $\mathbf{h}_{t,k}^l = \left[\mathbf{h}_t, a_{t+k-1}^{l-1}, \mathbf{e}_k\right]$. We then encode each $\mathbf{h}_{t,k}^l$ with a shared MLP network and process them through GRU (Cho et al., 2014) to obtain $\mathbf{s}_{t,k}^l = f_\theta^{\texttt{GRU}}(f_\theta^{\texttt{MLP}}(\mathbf{h}_{t,1}^l), ..., f_\theta^{\texttt{MLP}}(\mathbf{h}_{t,k}^l))$. We find that this design empirically performs better than directly giving actions as inputs to GRU. We then use a shared projection layer to map each $\mathbf{s}_{t,k}^l$ into Q-values at each sequence step $k$, *i.e.,* $[Q_\theta^{l,k}(\mathbf{h}_t, a_{t+k-1}^{l,b}, a_{t:t+K}^{l-1})]_{b=1}^B = f_\theta^{\texttt{proj}}(\mathbf{s}_{t,k}^l)$.

### 3.2   Action Execution and Training Details

**Executing action with temporal ensemble**    With the policy that outputs an action sequence $a_{t:t+K}$, one question is how to execute actions at time step $i \in \{t, ..., t + K - 1\}$. For this, we use *temporal ensemble* (Zhao et al., 2023) that computes $a_{t:t+K}$ every time step, saves it to a buffer, and executes a weighted average $\sum_i w_i \bar{a}_t^i / \sum w_i$ where $\bar{a}_t^i$ denotes an action for step $t$ computed at step $t - i$, $w_i = \exp(-m * i)$ denotes a weight that assigns higher value to more recent actions, with $m$ as a hyperparameter that adjusts the weighting magnitude. We find this scheme outperforms the alternative of computing $a_{t:t+K}$ every $K$ steps and executing each action for subsequent $K$ steps on most tasks we considered, except on several tasks that need reactive control.

**Storing training data**    When storing samples from the environment, we store a transition $(\mathbf{o}_t, \hat{a}_t, r_{t+1}, \mathbf{o}_{t+1})$ where $\hat{a}_t$ denotes an action executed at time step $t$. For instance, if we use temporal ensemble for action execution, $\hat{a}_t$ is a weighted average of action outputs obtained from previous $K$ time steps, i.e., $\hat{a}_t = \sum_i w_i \bar{a}_t^i / \sum w_i$.

**Sampling training data from a replay buffer**    When sampling training data from the replay buffer, we sample a transition with action sequence, *i.e.,* $(\mathbf{o}_t, \hat{a}_{t:t+K}, r_{t+1}, \mathbf{o}_{t+1})$. If we sample time step $t$ near the end of episode so that we do not have enough data to construct a full action sequence, we fill the action sequence with *null* actions. In particular, in position control where we specify the position of joints or end effectors, we repeat the action from the last step so that the agent learns not to change the position. In torque control where we specify the force to apply, we set the action after the last step to zero so that agent learns to not to apply force.

## 4   Experiment

We study CQN-**AS** on 25 humanoid control tasks from BiGym (Chernyadev et al., 2024) and 20 tabletop manipulation tasks from RLBench (James et al., 2020) (see Figure 4 for examples of robotic tasks). To focus on challenging robotic tasks that aim to induce policies generating realistic behaviors, we consider a practical setup of demo-driven RL where we initialize training with a modest amount of expert demonstrations and then train with online data. In particular, our experiments are designed to investigate the following questions:

- Can CQN-**AS** quickly match the performance of a recent BC algorithm (Zhao et al., 2023) and surpass it through online learning? How does CQN-**AS** compare to previous model-free RL algorithms (Yarats et al., 2022; Seo et al., 2024)?
- What is the effect of each component in CQN-**AS**?
- Under which conditions is CQN-**AS** effective?

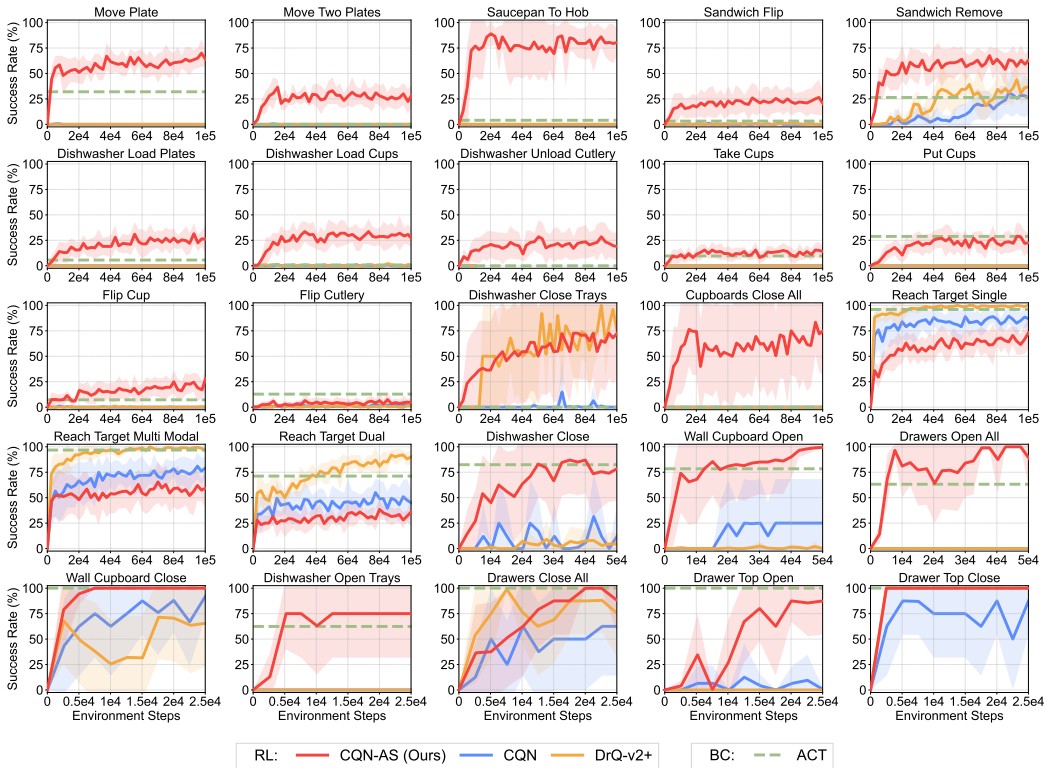

Figure 5: **BiGym results** on 25 sparsely-rewarded mobile bi-manual manipulation tasks. All RL algorithms are trained from scratch, with a replay buffer initialized with 17 to 60 *human-collected* demonstrations, and with an auxiliary BC objective. We report the success rate over 25 episodes. The solid line and shaded regions represent the mean and confidence intervals, respectively, across 8 runs.

**Baselines**    We consider model-free RL baselines that learn deterministic policies, as we find that stochastic policies struggle to solve fine-grained manipulation tasks. Specifically, we consider (i) Coarse-to-fine Q-Network (CQN; Seo et al. 2024), our backbone algorithm and (ii) DrQ-v2+, an optimized demo-driven variant of an actor-critic algorithm DrQ-v2 (Yarats et al., 2022) that uses a deterministic policy algorithm and data augmentation. We further consider (iii) Action Chunking Transformer (ACT; Zhao et al. 2023) that trains a transformer (Vaswani et al., 2017) policy to predict action sequences from expert demonstrations and utilizes temporal ensemble, as our BC baseline.

**Implementation details**    For training with expert demonstrations, we follow the setup of Seo et al. (2024). We keep a separate replay buffer that stores demonstrations and sample half of training data from demonstrations. We also relabel successful online episodes as demonstrations and store them in the demonstration replay buffer. For CQN-**AS**, we use an auxiliary BC loss from Seo et al. (2024) based on large margin loss (Hester et al., 2018). For actor-critic baselines, we use an auxiliary BC loss that minimizes L2 loss between the policy outputs and expert actions.

## 4.1    BiGym Experiments

We study CQN-**AS** on mobile bi-manual manipulation tasks from BiGym (Chernyadev et al., 2024). BiGym's *human-collected* demonstrations are often noisy and multi-modal, posing challenges for RL algorithms. These algorithms must effectively leverage the information within demonstrations to learn strong initial behaviors, thereby mitigating exploration difficulties in sparsely rewarded tasks.

**Setup**    We consider 25 BiGym tasks with 17 to 60 demonstrations[2]. We use RGB observations with 84×84 resolution from `head`, `left_wrist`, and `right_wrist` cameras. We also use low-dimensional proprioceptive states. We use (i) absolute joint position control action mode and (ii)

---

[2]BiGym benchmark provides different number of successful demonstrations for each task. But we use the same number of demonstrations for all algorithms. See Appendix B for more details.

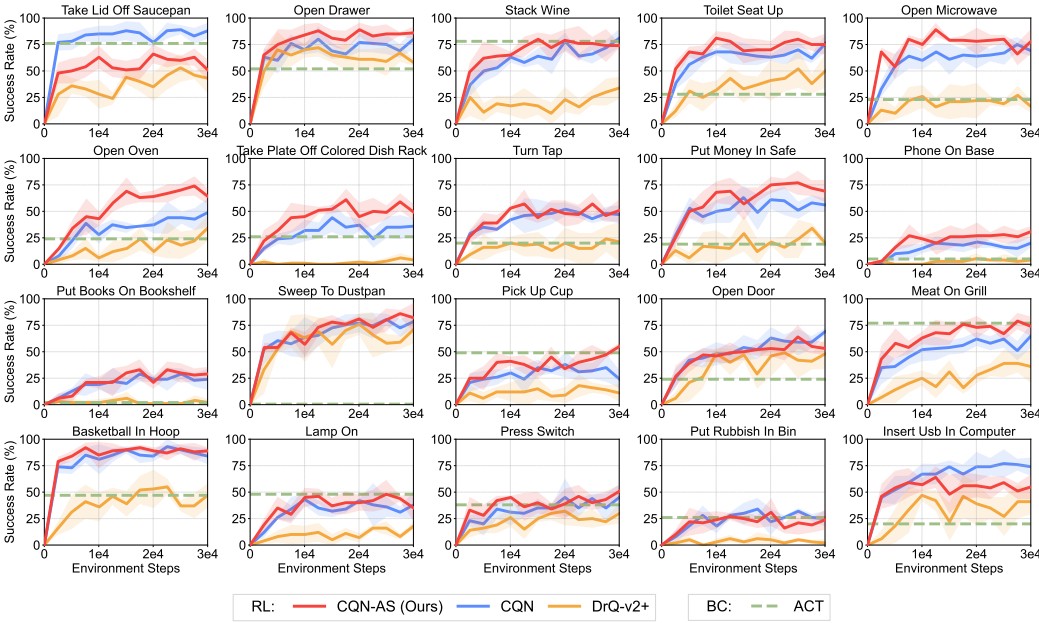

Figure 6: **RLBench results** on 20 sparsely-rewarded tabletop manipulation tasks from RLBench (James et al., 2020). All RL algorithms are trained from scratch, with a replay buffer initialized with 100 *synthetic* demonstrations generated via motion-planning, and with an auxiliary BC objective. *As expected*, with synthetic demonstrations, CQN-**AS** achieves similar performance to CQN on most tasks. However, CQN-**AS** often significantly outperforms baselines on several challenging, long-horizon tasks such as Open Oven. We report the success rate over 25 episodes. The solid line and shaded regions represent the mean and confidence intervals, respectively, across 4 runs.

floating base that replaces locomotion with classic controllers. We use the same set of hyperparameters for all the tasks. Details on BiGym experiments are available in Appendix B.

**Comparison to baselines** Figure 5 shows that CQN-**AS** quickly matches the performance of ACT and outperforms it through online learning on most tasks, while other RL algorithms fail to do so especially on challenging long-horizon tasks such as Move Plate and Saucepan To Hob. A notable result here is that CQN-**AS** *enables* solving challenging BiGym tasks while other RL baselines completely fail as they achieve a 0% success rate on many tasks.

**Limitation** However, CQN-**AS** struggles to achieve meaningful success rate on some of the long-horizon tasks that require interaction with delicate objects such as cups or cutlery. This leaves room for future work to incorporate advanced vision encoders (He et al., 2016) or critic architectures (Chebotar et al., 2023; Springenberg et al., 2024).

## 4.2 RLBench Experiments

We also study CQN-**AS** on manipulation tasks from RLBench (James et al., 2020). Unlike BiGym, RLBench provides synthetic demonstrations generated via motion planning, which are cleaner and more consistent. This allows us to examine whether CQN-**AS** is also effective in settings with *clean*, unambiguous demonstrations – where the effect of each single-step action is easier to interpret.

**Setup** We use the official CQN implementation for collecting demonstrations and reproducing the baseline results on the same set of tasks. We use RGB observations with 84×84 resolution from front, wrist, left_shoulder, and right_shoulder cameras. We also use low-dimensional proprioceptive states consisting of 7-dimensional joint positions and a binary value for gripper open. We use 100 demonstrations and delta joint position control action mode. We use the same set of hyperparameters for all the tasks, in particular, we use action sequence of length 4. More details on RLBench experiments are available in Appendix B.

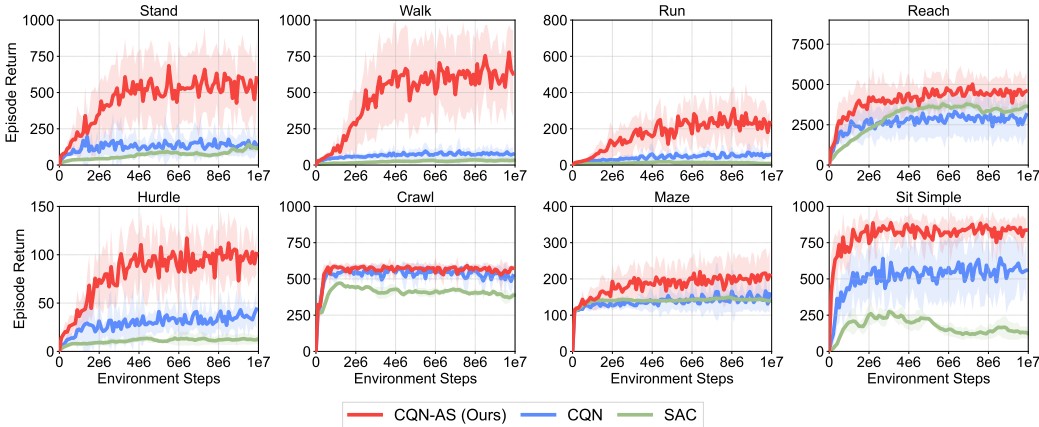

Figure 7: **HumanoidBench results** on eight densely-rewarded humanoid control tasks (Sferrazza et al., 2024). All the experiments start from scratch and all the methods do not have an auxiliary BC objective. CQN-**AS** significantly improves the performance of underlying RL algorithm CQN, while outperforming a model-free RL baseline, SAC. For CQN-**AS** and CQN, we report the results aggregated over 8 runs. For SAC, we report the results aggregated over 3 runs available from public website. The solid line and shaded regions represent the mean and confidence intervals.

**CQN-AS is also effective with *clean* demonstrations**   Because RLBench provides synthetic *clean* demonstrations, as we expected, Figure 6 shows that CQN-**AS** achieves *similar* performance to CQN on most tasks, except 2/25 tasks where it hurts the performance. But we still find that CQN-**AS** achieves quite superior performance to CQN on some challenging long-horizon tasks such as `Open Oven` or `Take Plate Off Colored Dish Rack`. These results show that CQN-**AS** can be used in various benchmark with different characteristics.

### 4.3   HumanoidBench Experiments

To show that CQN-**AS** is generally applicable to tasks without demonstrations, we also study CQN-**AS** on densely-rewarded tasks from HumanoidBench (Sferrazza et al., 2024).

**Setup**   We follow a standard setup that trains RL agents from scratch. We use low-dimensional states consisting of proprioception and privileged task information as inputs. For tasks, we simply select the first 8 locomotion tasks in the benchmark. For baselines, we consider CQN and Soft Actor-Critic (SAC) (Haarnoja et al., 2018). For SAC, we use the results available from HumanoidBench repository, which are evaluated on *tasks with dexterous hands*. For CQN-**AS** and CQN, we also evaluate them on tasks with hands. We use the same set of hyperparameters for all the tasks (see Appendix B).

**Comparison to baselines**   Figure 7 shows that, by learning the critic network with action sequence, CQN-**AS** outperforms other model-free RL baselines, *i.e.,* CQN and SAC, on most tasks. In particular, the difference between CQN-**AS** and baselines becomes larger as the task gets more difficult, *e.g.,* baselines fail to achieve high episode return on `Walk` and `Run` tasks but CQN-**AS** achieves strong performance. This result shows that our idea of using action sequence can be applicable to generic setup without demonstrations.

### 4.4   Ablation Studies, Analysis, Failure Cases

**Effect of action sequence length**   Figure 8a shows the performance of CQN-**AS** with different action sequence lengths. We find that training the critic network with longer action sequences tends to consistently improve performance, plateaus or decreases performance if the sequences get too long.

**RL objective is crucial for strong performance**   Figure 8b shows the performance of CQN-**AS** without RL objective that trains the model only with BC objective on successful demonstrations. We find this baseline significantly underperforms CQN-**AS**, which shows that RL objective enables the agent to learn from trial-and-error experiences.

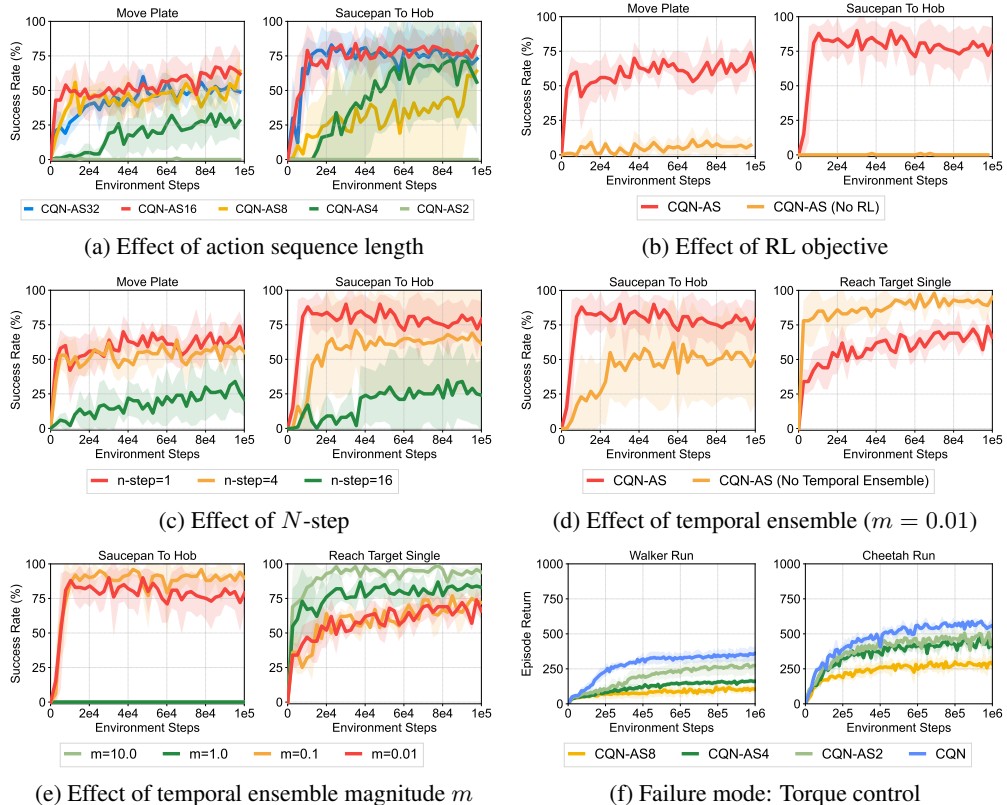

Figure 8: **Ablation studies and analysis** on the effect of (a) action sequence, (b) RL objective, (c) $N$-step return, and (d & e) temporal ensemble. (f) We also provide results on locomotion tasks from DeepMind Control Suite (Tassa et al., 2020), where CQN-**AS** fails to improve performance. The solid line and shaded regions represent the mean and confidence intervals, respectively, across 4 runs.

**Effect of $N$-step return**   Figure 8c shows experimental results with varying $N$-step returns. We find that too high $N$-step return significantly degrades performance. We hypothesize this is because the variance from $N$-step return makes it difficult to learn useful value functions.

**Effect of temporal ensemble**   Figure 8d shows that performance degrades without temporal ensemble on `Saucepan To Hob` as temporal ensemble induces a smooth motion and thus improves performance in fine-grained control tasks. But we also find that temporal ensemble is harmful on `Reach Target Single`. This is because temporal ensemble uses predictions from previous steps and thus makes it difficult to refine behaviors based on recent observations. Nonetheless, we use temporal ensemble for all the tasks as it helps on most tasks and we aim to use the same set of hyperparameters.

**Effect of temporal ensemble magnitude**   We further provide results with different temporal ensemble magnitudes by adjusting a hyperparameter $m$ in Figure 8e. Here, higher $m$ puts higher weights on recent actions and thus very high $m$ corresponds to using only first action. Similarly to previous paragraph, we find that higher $m$ leads to better performance on `Reach Target Single` that needs fast reaction, but degrades performance on `Saucepan To Hob`.

**Failure case: Torque control**   Figure 8f shows that CQN-**AS** underperforms CQN on locomotion tasks with torque control, which are drawn from the DeepMind Control Suite (Tassa et al., 2020). We hypothesize that this performance degradation arises because sequences of joint positions tend to have clearer semantic structure in joint space, making them easier to learn from compared to sequences of raw torques. Addressing this failure case represents an interesting direction for future work.

# 5 Related Work

**Behavior cloning with action sequence**   Recent behavior cloning approaches have shown that predicting a sequence of actions enables the policy to imitate noisy expert trajectories and helps in dealing with idle actions from human pauses during data collection (Zhao et al., 2023; Chi et al., 2023). Notably, Zhao et al. (2023) train a transformer model (Vaswani et al., 2017) that predicts action sequence and Chi et al. (2023) train a denoising diffusion model (Ho et al., 2020) that approximates the action distributions. This idea has been extended to multi-task setup (Bharadhwaj et al., 2024), mobile manipulation (Fu et al., 2024b) and humanoid control (Fu et al., 2024a). Our work is inspired by this line of research and proposes to learn RL agents with action sequence.

**Reinforcement learning with action sequence**   In the context of RL, Medini & Shrivastava (2019) propose to pre-compute frequent action sequences from expert demonstrations and augment the action space with these sequences. However, this idea introduces additional complexity and is not scalable to setups without demonstrations. One recent work relevant to ours is Saanum et al. (2024) that encourage a sequence of actions from RL agents to be predictable and smooth. Our work differs in that we directly incorporate action sequences into value learning. Recently, Ankile et al. (2024) point out that RL with action sequence is challenging and instead propose to use RL for learning a single-step policy that corrects action sequence predictions from BC. In contrast, we show that RL with action sequence is feasible and improves performance of RL algorithms.

**Multi-token prediction**   Recent large language models have incorporated a notably similar idea to predicting action sequences from demonstrations – predicting multiple future tokens at once, or multi-token prediction (Gloeckle et al., 2024; Liu et al., 2024). For instance, Gloeckle et al. (2024) show that predicting multiple $n$ future tokens in parallel with $n$ independent output heads improves the performance and can accelerate inference speed. DeepSeek-V3 (Liu et al., 2024) also make a similar observation but with a sequential multi-token prediction. It would be interesting to see whether our idea can be utilized for fine-tuning these models with multi-token prediction.

**Hierarchical reinforcement learning**   Approaches that learn RL agents with temporally extended high-level actions, or options, have been well studied (Sutton, 1988). The key idea is to train high-level policies that output options by manually defining subgoals (Kulkarni et al., 2016; Dayan & Hinton, 1992) or learning options from data (Bacon et al., 2017; Vezhnevets et al., 2017; Nachum et al., 2018), and then train a low-level agent that learns to execute low-level actions conditioned on options. Our work is not directly comparable to these works as we do not abstract temporally extended actions but use raw action sequences for value learning.

# 6 Discussion

We have presented Coarse-to-fine Q-Network with **A**ction **S**equence (CQN-**AS**), a critic-only RL algorithm that trains a critic network to output Q-values over action sequences. Extensive experiments in benchmarks with various setups show that our idea not only improves the performance of the base algorithm but also allows for solving complex tasks where prior RL algorithms fail.

**Limitations and future work**   One limitation of our work is the lack of real-world robot evaluation. Moreover, as discussed in Section 4.1 and Section 4.4, solving tasks involving small objects remains a limitation of our approach. One potential approach would be using strong pre-trained vision encoders, but we find that computational cost is often prohibitively large, which remains as an open problem. We are excited about future directions, including real-world RL with humanoid robots, incorporating advanced critic architectures (Kapturowski et al., 2023; Chebotar et al., 2023; Springenberg et al., 2024), bootstrapping RL agents from imitation learning (Hu et al., 2023; Xing et al., 2024) or offline RL (Nair et al., 2020; Lee et al., 2021), extending the idea to recent model-based RL approaches (Hafner et al., 2023; Hansen et al., 2024), extend parallel value learning scheme to autoregressive, multi-step Q-learning scheme (Kahn et al., 2018), fine-tuning vision-language-action models that use action sequence (Team et al., 2024; Doshi et al., 2024) or language models that use multi-token prediction (Gloeckle et al., 2024; Liu et al., 2024) with our algorithm, to name but a few.

## Acknowledgements

We thank Stephen James and Richie Lo for the discussion on the initial idea of this project. This work was supported in part by Multidisciplinary University Research Initiative (MURI) award by the Army Research Office (ARO) grant No. W911NF-23-1-0277. Pieter Abbeel holds concurrent appointments as a Professor at UC Berkeley and as an Amazon Scholar. This paper describes work performed at UC Berkeley and is not associated with Amazon.

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

## A  Motivating Experiments

**Additional details**    For Figure 2a, we use the same demonstrations used in our main experiments (see Appendix B for more details). For SAC and TD3 experiments with action sequences in Figure 2b, we implemented our code based on the official HumanoidBench repository. We use the hyperparameters in the repository for training SAC agents. For TD3, we use the standard deviation of 0.2 for exploration. We report the average target Q-values recorded throughout experiments. For Figure 2c, we train SAC and CQN agents with 6 original actions and 294 no-op actions with [-1, 1] action bounds and use an environment wrapper that slices out no-op actions.

**Experiment with 2D Point-mass environment**    To further motivate the use of action sequence for value learning, we train DQN agents (Mnih et al., 2015) on 2D Point-mass environment with discrete action spaces. In particular, we train a `Raw` agent that trains with the raw discrete action space consisting of single-step accelerations parameterized by 8 discrete headings (cardinal and discrete directions) and 1 magnitude level, resulting in 8 total actions. We compare this against a `Sequence` agent that trains with the discrete action space that consists of smooth 5-step acceleration sequences parameterized by cubic Bezier curves instead of single-step accelerations. Both agents operate on a 2D double-integrator environment where the goal is to reach a target position. In Figure 9, as expected, we find that training DQN agent with pre-defined action sequences lead to faster convergence. Our main experimental results in Section 4 further show that, CQN-AS can achieve similar benefit of using action sequences without pre-defined set of action sequences on various challenging continuous control benchmarks.

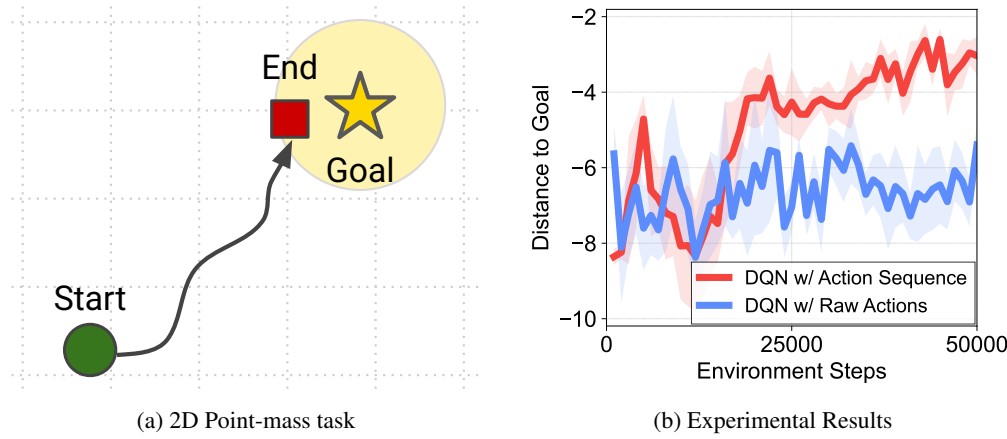

(a) 2D Point-mass task                    (b) Experimental Results

Figure 9: **2D Point-mass experiments** (a) We consider a simple toy environment where the goal is to reach the region around a randomly spawned goal point. (b) We show that, as expected, training with pre-defined action sequences lead to faster convergence. The solid line and shaded regions represent the mean and confidence intervals, respectively, across 10 runs.

# B  Experimental Details

**BiGym**  BiGym[3] (Chernyadev et al., 2024) is built upon MuJoCo (Todorov et al., 2012). We use Unitree H1 with two parallel grippers. We find that demonstrations available in the recent version of BiGym are not all successful. Therefore we adopt the strategy of replaying all the demonstrations and only use the successful ones as demonstrations. instead of discarding the failed demonstrations, we store them in a replay buffer as failure experiences. To avoid training with too few demonstrations, we exclude the tasks where the ratio of successful demonstrations is below 50%. Table 1 shows the list of 25 sparsely-rewarded mobile bi-manual manipulation tasks used in our experiments.

Table 1: **BiGym tasks** with their maximum episode length and number of successful demonstrations.

| Task | Length | Demos | Task | Length | Demos |
|------|--------|-------|------|--------|-------|
| Move Plate | 300 | 51 | Cupboards Close All | 620 | 53 |
| Move Two Plates | 550 | 30 | Reach Target Single | 100 | 30 |
| Saucepan To Hob | 440 | 28 | Reach Target Multi Modal | 100 | 60 |
| Sandwich Flip | 620 | 34 | Reach Target Dual | 100 | 50 |
| Sandwich Remove | 540 | 24 | Dishwasher Close | 375 | 44 |
| Dishwasher Load Plates | 560 | 17 | Wall Cupboard Open | 300 | 44 |
| Dishwasher Load Cups | 750 | 58 | Drawers Open All | 480 | 45 |
| Dishwasher Unload Cutlery | 620 | 29 | Wall Cupboard Close | 300 | 60 |
| Take Cups | 420 | 32 | Dishwasher Open Trays | 380 | 57 |
| Put Cups | 425 | 43 | Drawers Close All | 200 | 59 |
| Flip Cup | 550 | 45 | Drawer Top Open | 200 | 40 |
| Flip Cutlery | 500 | 43 | Drawer Top Close | 120 | 51 |
| Dishwasher Close Trays | 320 | 62 | | | |

**HumanoidBench**  HumanoidBench[4] (Sferrazza et al., 2024) is built upon MuJoCo (Todorov et al., 2012). We use Unitree H1 with two dexterous hands. We consider the first 8 locomotion tasks in the benchmark: `Stand`, `Walk`, `Run`, `Reach`, `Hurdle`, `Crawl`, `Maze`, `Sit Simple`. We use proprioceptive states and privileged task information instead of visual observations. Unlike BiGym and RLBench experiments, we do not utilize dueling network (Wang et al., 2016) and distributional critic (Bellemare et al., 2017) in HumanoidBench for faster experimentation.

**RLBench**  RLBench[5] (James et al., 2020) is built upon CoppeliaSim (Rohmer et al., 2013) and PyRep (James et al., 2019). We use a 7-DoF Franka Panda robot arm and a parallel gripper. Following the setup of Seo et al. (2024), we increase the velocity and acceleration of the arm by 2 times. For all experiments, we use 100 demonstrations generated via motion-planning. Table 2 shows the list of 20 sparsely-rewarded visual manipulation tasks used in our experiments.

Table 2: **RLBench tasks** with their maximum episode length used in our experiments.

| Task | Length | Task | Length |
|------|--------|------|--------|
| Take Lid Off Saucepan | 100 | Put Books On Bookshelf | 175 |
| Open Drawer | 100 | Sweep To Dustpan | 100 |
| Stack Wine | 150 | Pick Up Cup | 100 |
| Toilet Seat Up | 150 | Open Door | 125 |
| Open Microwave | 125 | Meat On Grill | 150 |
| Open Oven | 225 | Basketball In Hoop | 125 |
| Take Plate Off Colored Dish Rack | 150 | Lamp On | 100 |
| Turn Tap | 125 | Press Switch | 100 |
| Put Money In Safe | 150 | Put Rubbish In Bin | 150 |
| Phone on Base | 175 | Insert Usb In Computer | 100 |

---

[3] https://github.com/chernyadev/bigym
[4] https://github.com/carlosferrazza/humanoid-bench
[5] https://github.com/stepjam/RLBench

**Hyperparameters** We use the same set of hyperparameters across the tasks in each domain. For hyperparameters shared across CQN and CQN-**AS**, we use the same hyperparameters for both algorithms for a fair comparison. We provide detailed hyperparameters for BiGym and RLBench experiments in Table 3 and HumanoidBench experiments in Table 4

Table 3: Hyperparameters for demo-driven vision-based experiments in BiGym and RLBench

| Hyperparameter | Value |
|---|---|
| Image resolution | $84 \times 84 \times 3$ |
| Image augmentation | RandomShift (Yarats et al., 2022) |
| Frame stack | 4 (BiGym) / 8 (RLBench) |
| CNN - Architecture | Conv (c=[32, 64, 128, 256], s=2, p=1) |
| MLP - Architecture | Linear (c=[512, 512, 64, 512, 512], bias=False) (BiGym) |
| | Linear (c=[64, 512, 512], bias=False) (RLBench) |
| CNN & MLP - Activation | SiLU (Hendrycks & Gimpel, 2016) and LayerNorm (Ba et al., 2016) |
| GRU - Architecture | GRU (c=[512], bidirectional=False) |
| Dueling network | True |
| C51 - Atoms | 51 |
| C51 - $v_{min}$, $v_{max}$ | -2, 2 |
| Action sequence | 16 (BiGym) / 4 (RLBench) |
| Temporal ensemble weight $m$ | 0.01 |
| Levels | 3 |
| Bins | 5 |
| BC loss ($\mathcal{L}_{BC}$) scale | 1.0 |
| RL loss ($\mathcal{L}_{RL}$) scale | 0.1 |
| Relabeling as demonstrations | True |
| Data-driven action scaling | True |
| Action mode | Absolute Joint (BiGym), Delta Joint (RLBench) |
| Exploration noise | $\epsilon \sim \mathcal{N}(0, 0.01)$ |
| Target critic update ratio ($\tau$) | 0.02 |
| N-step return | 1 |
| Batch size | 256 |
| Demo batch size | 256 |
| Optimizer | AdamW (Loshchilov & Hutter, 2019) |
| Learning rate | 5e-5 |
| Weight decay | 0.1 |

**Computing hardware** For BiGym and Humanoid experiments, we use NVIDIA A5000 GPU with 24GB VRAM. With A5000, each BiGym experiment with 100K environment steps take 16 hours, and each HumanoidBench experiment with 10M environment steps take 40 hours. For RLBench experiments, we use NVIDIA RTX 2080Ti GPU, with which each experiment with 30K environment steps take 6.5 hours. We find that CQN-AS takes 13% more memory compared to CQN and is 40% slower than CQN. Overall, CQN-**AS** is around 33% slower than running CQN because larger architecture slows down both training and inference.

**Baseline implementation** For CQN (Seo et al., 2024) and DrQ-v2+ (Yarats et al., 2022), we use the implementation available from the official CQN implementation[6]. For ACT (Zhao et al., 2023), we use the implementation from RoboBase repository[7]. For SAC (Haarnoja et al., 2018), DreamerV3 (Hafner et al., 2023), and TD-MPC2 (Hansen et al., 2024), we use results provided in HumanoidBench[8] repository (Sferrazza et al., 2024).

---

[6]https://github.com/younggyoseo/CQN
[7]https://github.com/robobase-org/robobase
[8]https://github.com/carlosferrazza/humanoid-bench

Table 4: Hyperparameters for state-based experiments in HumanoidBench

| Hyperparameter | Value |
| --- | --- |
| MLP - Architecture | Linear (c=[512, 512], bias=False) |
| CNN & MLP - Activation | SiLU (Hendrycks & Gimpel, 2016) and LayerNorm (Ba et al., 2016) |
| GRU - Architecture | GRU (c=[512], bidirectional=False) |
| Dueling network | True |
| Action sequence | 4 |
| Temporal ensemble weight $m$ | 0.01 |
| Levels | 3 |
| Bins | 5 |
| RL loss ($\mathcal{L}_{\text{RL}}$) scale | 1.0 |
| Action mode | Absolute Joint |
| Exploration noise | $\epsilon \sim \mathcal{N}(0, 0.01)$ |
| Target critic update ratio ($\tau$) | 1.0 |
| Target critic update interval ($\tau$) | 100 |
| Update-to-data ratio (UTD) | 0.5 |
| N-step return | 3 |
| Batch size | 128 |
| Optimizer | AdamW (Loshchilov & Hutter, 2019) |
| Learning rate | 5e-5 |
| Weight decay | 0.1 |

# C  Full description of CQN and CQN-AS

This section provides the formulation of CQN and CQN-**AS** with $n$-dimensional actions.

## C.1  Coarse-to-fine Q-Network

Let $a_t^{l,n}$ be an action at level $l$ and dimension $n$ and $\mathbf{a}_t^l = \{a_t^{l,1}, ..., a_t^{l,N}\}$ be actions at level $l$ with $\mathbf{a}_t^0$ being zero vector. We then define coarse-to-fine critic to consist of multiple Q-networks:

$$Q_\theta^{l,n}(\mathbf{h}_t, a_t^{l,n}, \mathbf{a}_t^{l-1}) = \left[Q_\theta^l(\mathbf{h}_t, a_t^{l,n} = a_t^{l,n,b}, \mathbf{a}_t^{l-1})\right]_{b=1}^{B} \quad \text{for } l \in \{1, ..., L\} \text{ and } n \in \{1, ..., N\} \tag{3}$$

Where $B$ denotes the number of bins. We optimize the critic network with the following objective:

$$\sum_n \sum_l \left(Q_\theta^{l,n}(\mathbf{h}_t, a_t^{l,n}, \mathbf{a}_t^{l-1}) - r_{t+1} - \gamma \max_{a'} Q_{\bar\theta}^{l,n}(\mathbf{h}_{t+1}, a', \pi^{l-1}(\mathbf{h}_{t+1}))\right)^2, \tag{4}$$

where $\bar\theta$ are delayed parameters for a target network (Polyak & Juditsky, 1992) and $\pi^l$ is a policy that outputs the action $\mathbf{a}_t^l$ at each level $l$ via the inference steps with our critic, *i.e.,* $\pi^l(\mathbf{h}_t) = \mathbf{a}_t^l$.

**Action inference**  To output actions at time step $t$ with the critic, CQN first initializes constants $a_t^{n,\texttt{low}}$ and $a_t^{n,\texttt{high}}$ with $-1$ and $1$ for each $n$. Then the following steps are repeated for $l \in \{1, ..., L\}$:

- Step 1 (Discretization): Discretize an interval $[a_t^{n,\texttt{low}}, a_t^{n,\texttt{high}}]$ into $B$ uniform intervals, and each of these intervals become an action space for $Q_\theta^{l,n}$.

- Step 2 (Bin selection): Find the bin with the highest Q-value, set $a_t^{l,n}$ to the centroid of the selected bin, and aggregate actions from all dimensions to $\mathbf{a}_t^l$.

- Step 3 (Zoom-in): Set $a_t^{n,\texttt{low}}$ and $a_t^{n,\texttt{high}}$ to the minimum and maximum of the selected bin, which intuitively can be seen as zooming-into each bin.

We then use the last level's action $\mathbf{a}_t^L$ as the action at time step $t$.

**Computing Q-values**  To compute Q-values for given actions $\mathbf{a}_t$, CQN first initializes constants $a_t^{n,\texttt{low}}$ and $a_t^{n,\texttt{high}}$ with $-1$ and $1$ for each $n$. We then repeat the following steps for $l \in \{1, ..., L\}$:

- Step 1 (Discretization): Discretize an interval $[a_t^{n,\texttt{low}}, a_t^{n,\texttt{high}}]$ into $B$ uniform intervals, and each of these intervals become an action space for $Q_\theta^{l,n}$.

- Step 2 (Bin selection): Find the bin that contains input action $\mathbf{a}_t$, compute $a_t^{l,n}$ for the selected interval, and compute Q-values $Q_\theta^{l,n}(\mathbf{h}_t, a_t^{l,n}, \mathbf{a}_t^{l-1})$.

- Step 3 (Zoom-in): Set $a_t^{n,\texttt{low}}$ and $a_t^{n,\texttt{high}}$ to the minimum and maximum of the selected bin, which intuitively can be seen as zooming-into each bin.

We then use a set of Q-values $\{Q_\theta^{l,n}(\mathbf{h}_t, a_t^{l,n}, \mathbf{a}_t^{l-1})\}_{l=1}^L$ for given actions $\mathbf{a}_t$.

## C.2  Coarse-to-fine Critic with Action Sequence

Let $\mathbf{a}_{t:t+K}^l = \{\mathbf{a}_t^l, ..., \mathbf{a}_{t+K-1}^l\}$ be an action sequence at level $l$ and $\mathbf{a}_{t:t+K}^0$ be zero vector. Our critic network consists of multiple Q-networks for each level $l$, dimension $n$, and sequence step $k$:

$$Q_\theta^{l,n,k}(\mathbf{h}_t, a_{t+k-1}^{l,n}, \mathbf{a}_{t:t+K}^{l-1}) = \left[Q_\theta^{l,n,k}(\mathbf{h}_t, a_{t+k-1}^{l,n} = a_{t+k-1}^{l,n,b}, \mathbf{a}_{t:t+K}^{l-1})\right]_{b=1}^B$$
$$\text{for } l \in \{1, ..., L\}, \ n \in \{1, ..., N\} \text{ and } k \in \{1, ..., K\} \tag{5}$$

We optimize the critic network with the following objective:

$$\sum_n \sum_l \sum_k \left(Q_\theta^{l,n,k}(\mathbf{h}_t, a_t^{l,n}, \mathbf{a}_{t:t+K}^{l-1}) - r_{t+1} - \gamma \max_{a'} Q_{\bar\theta}^{l,n,k}(\mathbf{h}_{t+1}, a', \pi_K^{l-1}(\mathbf{h}_{t+1}))\right)^2, \tag{6}$$

where $\pi_K^l$ is an action sequence policy that outputs the action sequence $\mathbf{a}_{t:t+K}^l$. In practice, we compute Q-values for all sequence step $k \in \{1, ..., K\}$ and all action dimension $n \in \{1, ..., N\}$ in parallel. This can be seen as extending the idea of Seyde et al. (2023), which learns decentralized Q-networks for action dimensions, into action sequence dimension. As we mentioned in Section 3.1, we find this simple scheme works well on challenging tasks with high-dimensional action spaces.

**Architecture**  Let $\mathbf{e}_k$ denote an one-hot encoding for $k$. For each level $l$, we construct features for each sequence step $k$ as $\mathbf{h}_{t,k}^l = [\mathbf{h}_t, \mathbf{a}_{t+k-1}^{l-1}, \mathbf{e}_k]$. We encode each $\mathbf{h}_{t,k}^l$ with a shared MLP network and process them through GRU (Cho et al., 2014) to obtain $\mathbf{s}_{t,k}^l = f_\theta^{\texttt{GRU}}(f_\theta^{\texttt{MLP}}(\mathbf{h}_{t,1}^l), ..., f_\theta^{\texttt{MLP}}(\mathbf{h}_{t,k}^l))$. We use a shared projection layer to map each $\mathbf{s}_{t,k}^l$ into Q-values at each sequence step $k$, *i.e.*, $f_\theta^{\texttt{proj}}(\mathbf{s}_{t,k}^l) = \{[Q_\theta^{l,k}(\mathbf{h}_t, a_{t+k-1}^{l,n,b}, \mathbf{a}_{t:t+K}^{l-1})]_{b=1}^B\}_{n=1}^N$. We compute Q-values for all dimensions $n \in \{1, ..., N\}$ at the same time with a big linear layer, which follows the design of Seo et al. (2024).

# D  Additional Preliminary Experiments

**Offline RL with CQN-AS**  To further investigate whether CQN-AS formulation is compatible with offline RL, we conduct preliminary experiments on BiGym's `Sandwich Remove` task. Specifically, we combine CQN-AS with Cal-QL (Nakamoto et al., 2023) and train it on the dataset that consists of 26 successful demonstrations and 10 failed trajectories. We find that CQN-AS + Cal-QL achieves 33 ($\pm$ 6.8) % while CQN + Cal-QL achieves 7 ($\pm$ 14) %, which shows CQN-AS can be indeed effective for offline setup. We leave further exhaustive investigation as an interesting future work.

**Experiments with ResNet-18**  To investigate if using larger and stronger pre-trained vision encoders such as ResNet (He et al., 2016) can improve performance, we tried running CQN-AS with ResNet-18 encoder on BiGym tasks. However, we find that it requires a GPU with at least 48GB memory and is extremely slow to train. We will leave this direction of incorporating larger vision encoders in an efficient manner as a future direction.

