# OpenReview forum: "Coarse-to-fine Q-Network with Action Sequence for Data-Efficient Reinforcement Learning"
_NeurIPS.cc/2025/Conference — NeurIPS 2025 poster_

### Official Review · Reviewer_yZtr · 2025-06-06

**Clarity:** 3
**Significance:** 3
**Originality:** 3
**Rating:** 4
**Confidence:** 4

**Summary:**

Following a recent trend in behavioral cloning for robot learning, the authors augment an existing reinforcement learning (RL) algorithm, Coarse-to-Fine Q-Network (CQN) with the capability to deal with action sequences. They evaluate their algorithm, CQN-AS, on 3 sets of benchmarks (BiGym, RLBench and HumanoidBench) and show that CQN-AS generally outperforms several RL baselines which do not play action sequences and behavioral cloning (BC) baselines that do. They also provide an ablation study.

**Questions:**

*Major issues:*

- In the introduction, the authors outline that actor-critic algorithms dealing with action sequences suffer from over-estimation bias. Then they advocate for critic-only algorithms with the argument that "there is no separate actor that may exploit value functions" and they pretend that as their contribution, CQN-AS, is based on a critic-only approach, it will avoid the value overestimation problem. First, the argument is wrong: DQN is a critic-only algorithm (the most basic one) and it suffers from over-estimation bias, as the mitigation with Double DQN clearly shows. Second, the authors do not measure over-estimation bias in CQN-AS in the empirical study, so they do not show that it mitigates over-estimation bias. A study of the role of over-estimation bias in RL algorithms dealing with action sequences could be a useful contribution of this paper...

- About the N-step return, lines 112-117 the authors suggest that using from 1 to 3 steps works the best, but their empirical study only provides results with 1 (i.e. not N-step return) and 4 steps, which performs worse, hence suggesting that one should avoid using N-step return. This inconsistency must be solved, either by providing results with 2 and 3 steps, or by removing the N-step return idea from the main paper (showing that it fails in some Appendix may help readers avoid to try this again).

- In Figures 5 and 6, the performance of all algorithms including CQN-AS always starts at 0, though the policy is pre-trained with a few (17 to 60) expert demonstrations. Do the figures start before training with the demonstrations? Otherwise, could the authors explain why?

- Did all the baselines benefit from the same 17 to 60 expert demonstrations pre-training process? I assume it is the case for CQN, but for DrQ-v2+, it is less obvious. Otherwise I'm afraid the comparison is not scientifically valid.

- I don't find the message from the comparison between *human-collected* and *synthetic (clean)* demonstrations (BiGym vs RLBench experiments) convincing enough. The authors do not provide compelling evidence that this feature is playing a key role, it could be other factors in the different benchmarks that explain the difference. A controlled experiment where using synthetic versus human-collected data is the only changing factor (e.g. same benchmark, just changing the dataset) would be necessary to evaluate the role of this feature. Besides, as it only plays a role in pre-training and then balanced sampling, it is unclear why it should make such a difference.

- What are the differences between CQN-AS with the RL part ablated and BC-ACT? If they are close to be the same, I don't find Fig. 7b useful, as Fig. 5 already shows more or less the same results.

*More local points:*

- The authors could replace Figures 5 and 6 with performance profiles (see the Rliable library and "Reinforcement learning at the edge of the statistical precipice", Agrawal et al. NeurIPS 2021) and reject the full set of curves to an Appendix. The saved space could be used to include HumanoidBench results into the main paper
- The authors seem to be using balanced sampling of expert demonstrations, they should give a ref to this approach (Ross&Bagnell 2012 or the more recent RLPD paper)
- Did the authors try to ablate expert pre-training?
- On the effect of action length, why didn't the authors try longer sequences (32, 64, etc.) as they found that the longer, the better? Using 50 is a common practice in Diffusion policy based BC...
- The studies of the effect of temporal ensemble and of its magnitude (Figs. 7d and 7e) could be merged into a single study and the value of m is not given in Fig 7d.

- In the conclusion: "One limitation of our work is the lack of real-world robot evaluation,as RL with fragile humanoid hardware remains a significant challenge" -> Well, this is not the right argument, as some robots are not fragile humanoids :)

**Ethical Concerns:**

["NO or VERY MINOR ethics concerns only"]

**Final Justification:**

The authors did a good job in clarifying their points during the rebuttal, I'm now more convinced of the value of the paper, so I switched from a brderline reject to a borderline accpet.

**Limitations:**

The authors do not discuss the computational aspects of CQN-AS wrt CQN: is it faster or slower to train? Faster or slower at inference time? Does it require a larger memory? All these aspects will be important when switching to real robots

**Paper Formatting Concerns:**

no concern

**Quality:**

2

**Strengths And Weaknesses:**

Strengths:
- The paper fills a gap: it seems that the benefits of using action sequences in RL has not been shown before
- The paper is rather clear
- The empirical study uses a large set of benchmarks
- Results are generally good

Weaknesses:
- several points need further clarification (see below)
- we could expect a larger effort on understanding WHY action sequences help. Despite ablations, the paper does not provide empirical clues on that. Maybe using much simpler environments coudl help.
- the idea that the effect has to do with over-estimation bias is not supported (see below)
- an empirical comparison to (Saanum et al. 2024) would improve the paper
- the title is inadequate as there is no robot in this work

---

> ### Author Rebuttal · Authors · 2025-07-30
>
> Dear Reviewer yZtr,
>
> We really appreciate your time and effort for reviewing our paper. We provide our responses below.
>
> ---
>
> **Q1.  Empirical evidence that shows why using action sequences is helpful.**
>
> **A1.** In Introduction, we hypothesized that using action sequences can be helpful for training RL agents by making it easier to learn value functions. Our initial observation showed that using action sequences is helpful for predicting return-to-go from robotic demonstrations. We also showed that CQN-AS, which uses action sequences, achieves strong performance, directly supporting our hypothesis. Here, the key insight is that CQN-AS is a *critic-only algorithm* where actions are selected greedily based on the learned Q-values. The strong performance of CQN-AS is therefore a direct consequence of the critic learning a more accurate value function. We will further clarify this in the final version.
>
> ---
>
> **Q2. DQN also suffers from over-estimation bias. Further study on the effect of action sequences on over-estimation in RL algorithms could be a useful contribution**
>
> **A2. ** As you mentioned, RL algorithms with discrete actions also suffer from over-estimation bias as well. However, we would like to clarify that our focus in this work is on the over-estimation problem that happens when using high-dimensional actions as inputs to the critic network.
>
> In particular, what we find in this paper is that a combination of (i) a critic network that takes high-dimensional action sequences as inputs and (ii) the actor that is trained to output actions using gradients from the critic leads to severe value overestimation problems. We hypothesize this is because the critic network tends to have more function approximation errors from high-dimensional actions and the actor can easily exploit this error. On the other hand, critic-only algorithms can avoid this problem, leading to stable training with high-dimensional action sequences.
>
> To further support our point, we conducted additional toy experiments where we introduced redundant no-op actions for training TD3 and CQN agents on DMC Cheetah Run tasks. Specifically, we trained agents with 6 original actions and 294 no-op actions with [-1, 1] action bounds and used an environment wrapper that slices out no-op actions. In this setup, we found that CQN (critic-only algorithm with discrete actions) is indeed robust to the overestimation problem. In particular, we find that TD3 suffers from severe value overestimation with more no-op actions, while CQN with such no-op actions achieves similar performance with the original CQN agent. This clearly shows that a critic-only algorithm is more robust to training with high-dimensional action spaces.
>
> | Additional No-op actions | Episode Return (CQN) | Target Q-Values (CQN) | Episode Return (TD3) | Target Q-Values (TD3) |
> | :--- | :--- | :--- | :--- | :--- |
> | 0 | 236.61 | 18.84 | 140.67 | 16.25 |
> | 54 | 219.50 | 20.27 | 7.18 | -17.45 |
> | 144 | 185.38 | 16.42 | 0.56 | 1.00E+08 |
> | 294 | 202.94 | 18.66 | 0.27 | 4.00E+08 |
>
> Based on this observation, one potential alternative is to use expectile regression for Q-learning as in IQL [1], which can avoid value overestimation problems by avoiding the explicit maximization step in TD-learning. We will leave this as a future work and include relevant discussion in the final version.
>
> [1] Kostrikov, Ilya, Ashvin Nair, and Sergey Levine. "Offline reinforcement learning with implicit q-learning." International Conference on Learning Representations 2022
>
> ---
>
> **Q3. It is unclear why the properties of demonstrations should make such a difference.**
>
> **A3.** This is because our setup is a very challenging sparse-reward task setup, which makes it extremely difficult to solve the tasks unless we use demonstrations for guiding the training. This necessitates RL algorithms to effectively learn useful value functions from demonstrations and then use information from online trial-and-error experiences to further improve the performance. Therefore the properties of demonstrations become quite important -- if the characteristics of demonstrations (such as noisyness of trajectories) pose challenges in learning value functions from them, RL algorithms would suffer to learn initial meaningful behaviors and thus learn from online data as well in our setup.
>
> ---
>
> **Q4. Comparison between human-collected and synthetic demonstrations needs more controlled experiments.**
>
> **A4.** Thank you for your suggestion. To support our claim that value learning is easier with smoother and clean trajectories, we conducted preliminary experiments that train a return-to-go prediction model with smoothed trajectories from BiGym’s Move Plate task. In particular, we apply Gaussian filtering to demonstrations with different smoothing magnitudes, and train the model to predict return-to-go conditional on observation-action pairs. While these smoothed trajectories are not actually usable for policy learning as they have lost their precision, we conducted this experiment as a preliminary experiment for understanding the effect of demonstration characteristics on value learning. As shown in the below table, we find that the validation loss decreases with more smoothing, which shows that demonstration qualities can affect value learning, which thus affects RL training as we mentioned in **A3.**
>
> | Smoothing | Val Loss | Improvement |
> | :--- | :--- | :--- |
> | Baseline (no smoothing) | 0.0211 | |
> | Gaussian (sigma=0.5) | 0.0203 | 3.84\% |
> | Gaussian (sigma=1.0) | 0.0188 | 10.78\% |
> | Gaussian (sigma=1.5) | 0.0185 | 12.37\% |
> | Gaussian (sigma=2.0) | 0.0185 | 12.22\% |
> | Gaussian (sigma=2.5) | 0.0183 | 13.34\% |
>
> ---
>
> **Q5. The title is inadequate as there is no robot in this work.**
>
> **A5**. We respectfully disagree with the assessment of the title. Our use of simulation is in line with the standard and widely accepted practice in robot learning research, where algorithms are developed and validated for robotic agents.
>
> ---
>
> **Q6. Why do performances start from 0 despite pre-training with expert demonstrations?**
>
> **A6.**  We would like to clarify that there is no pre-training. We just initialize the training by initializing the replay buffer with expert demonstrations (and maintaining a separate demonstration replay buffer as well). Therefore all the methods’ performance starts from 0. We agree that this is confusing and we missed the explanation on this. We will include sentences to clarify our setup.
>
> ---
>
> **Q7. Did all baselines use the same 17 to 60 expert demonstrations?**
>
> **A7.** Yes. All methods use the same number of demonstrations. As we’ve written in Appendix C, we used different number of demonstrations for *different tasks*, but we used the same number of demonstrations for all algorithms within each task. We will clarify this in the final version.
>
> ---
>
> **Q8. Differences between CQN-AS without RL and ACT?**
>
> **A8.** As we’ve written in line 163-165, ACT is an Action Chunking Transformer [3] that trains a transformer model to output action sequences. On the other hand, CQN-AS without RL shares the same architecture with CQN-AS and also the discrete action space as well, which allows us to ablate the effect of RL objective.
>
> [2] Zhao, Tony Z., et al. "Learning fine-grained bimanual manipulation with low-cost hardware." Robotics: Science and Systems 2023
>
> ---
>
> **Q9. Inconsistency with regard to N-step return**
>
> **A9.** This is a good point. We wanted to emphasize that there is no need to set $N$ to be equal to the length of the action sequence ($K$), as that will train RL algorithms with very large $N$-step returns leading to unstable training. Our experiments find that the $N$-step of 1 or 4 (3 was a typo, we will fix this) works better than very large $N$ such as 16. But it looks like our intent was not clear. We will further clarify this and will move it to Appendix per your suggestion.
>
> ---
>
> **Q10. An empirical comparison to (Saanum et al. 2024) would improve the paper.**
>
> **A10.** Thank you for the suggestion. While we agree that such comparison would improve our paper, but it is quite non-trivial to compare our work to [Saanum et al., 2024] considering that (i) it has never been tested in our main setup that uses sparse-reward tasks and (ii) its implementation is not open-sourced.
>
> ---
>
> **Minor - Q1. Using Rliable library, adding references to balanced sampling, combining Figure 7(d) and (e)**
>
> **Minor - A1.** Thank you for your suggestions. We will incorporate them.
>
> ---
>
> **Minor - Q2. Performance with longer action sequences?**
>
> **Minor - A2.** We find that using longer action sequences ($K  > 16$) does not improve performance and sometimes degrade performance. We hypothesize this is because training with too long action sequences can be difficult, offsetting the benefit of learning with action sequences. We also note that the best action sequence length can be different for each dataset that has different action frequencies. We will include the relevant results and discussion in the final version.
>
> ---
>
> **Minor - Q3. Remarks on fragile humanoid hardware**
>
> **Minor - A3.** We will incorporate your point that not all robots are fragile.
>
> ---
>
> **Minor - Q4. Computational cost of CQN-AS compared to CQN is missing**
>
> **Minor - A4.** As we’ve written in Appendix C, training of CQN-AS is around 33% slower than CQN. At inference time, CQN-AS takes 13% more memory compared to CQN and is 40% slower than CQN. We believe these costs are definitely worthy considering the effectiveness of CQN-AS over CQN as shown in our BiGym experiments. Moreover, to further reduce the cost at inference time, we can consider adjusting the frequency of the temporal ensemble instead of computing action sequences at every time step. We will include the relevant discussion in the final version.

---

> > ### Comment · Reviewer_yZtr · 2025-08-01
> > **Several misunderstandings/unclarities need to be solved**
> >
> > When reading again my review and the authors' rebuttal, it seems that several misunderstandings need to be solved.
> >
> > - The authors summarize my first point as "(Is there) Empirical evidence that shows why using action sequences is helpful.", but my point is rather the following: "Is there empirical evidence that action sequences help because they mitigate over-estimation bias (rather than any other reason?)." Could the authors focus on this question?
> >
> > - the authors mention that they focus on the over-estimation problem that happens when using high-dimensional actions as inputs to the critic network (rather than general over-estimation), but since they discretize the action space, it is quite tautological that they remove the specific problem over-estimation problem raised by continuous actions. The question should rather be whether they mitigate over-estimation bias in general.
> >
> > - the experiment adding no-op actions is interesting, but can the authors explain why being robust to more actions means being robust to the overestimation problem?
> >
> > - about Q3 and A3, for the sake of a better understanding on why action sequences help, wouldn't it be better to investigate toy problems that can be analyzed more deeply and where confounding factors such as demonstration datasets etc. are not required?
> >
> > - about the title, I agree that from a machine learning point of view, working in simulation or on a real robot does not make a large difference, but from a robotics point of view it does. Doing RL with real robots is generally much harder than in simulation, roboticists may get attracted by your paper's title, try to apply your method and get disappointed. To avoid generting bad buzz in the robotics community, I would suggest replacing "robot" by "simulated robot" in the title (but I agree that you are not the only ML researchers to use this practice so I won't insist more).
> >
> > - About action sequence length, the paper says: "Figure 7a shows the performance of CQN-AS with different action sequence lengths. We find that training the critic network with longer action sequences tends to consistently improve performance.", which is less precise than your rebuttal statement that "We find that using longer action sequences ($K > 16$) does not improve performance and sometimes degrade performance." I think figure 7a should show a length over 16 and the conclusion should be updated.
> >
> > - Still about action sequence length, in the table you added to Q3/A3 of Reviewer 5gur, there is a pattern that the return for length 2 is lower than for length 1 and length 4. Any idea why?

---

> > > ### Author Response · Authors · 2025-08-02
> > >
> > > Dear Reviewer yZtr,
> > >
> > > We appreciate your prompt response and questions. We really hope that we successfully clarified your concerns with the following responses.
> > >
> > > ---
> > >
> > > **v2-Q1. Is there empirical evidence that action sequences help because they mitigate over-estimation bias (rather than any other reason?**
> > >
> > > **v2-A1.** Thank you for clarifying your point. We would like to clarify that (i) using action sequences cannot mitigate the overestimation problem -- actually we showed it is the cause of the value overestimation problem -- and  (ii) we did not intend to *address* the overestimation problem in general. Specifically, we wanted to build an RL method that uses action sequences, but we observed that using action sequences for RL suffers from a value overestimation problem (Please see **v2-A3** on why action sequences can cause this problem). Therefore we built our method upon the method that uses discrete action spaces -- which we found to avoid the issue of severe value overestimation and effectively utilize action sequences without suffering from training collapse.
> > >
> > > ---
> > >
> > > **v2-Q2. It is quite tautological that authors remove the specific over-estimation problem raised by continuous actions by using discrete action spaces. The question should rather be whether they mitigate over-estimation bias in general**
> > >
> > > **v2-A2.** We would like to clarify again that our goal is not in *addressing* the over-estimation problem when using high-dimensional actions as inputs, but in developing an RL method that can effectively utilize the action sequences. As you mentioned, and also as we mentioned in line 35 of the paper, we were able to *avoid* this problem by using discrete actions and showed that our proposed method indeed works with action sequences. With regard to why using discrete action spaces can be helpful in our context, please see **v2-A3**.
> > >
> > > ---
> > >
> > > **v2-Q3. The experiment adding no-op actions is interesting, but can the authors explain why being robust to more actions means being robust to the overestimation problem?**
> > >
> > > **v2-A3.** This is because it gets more difficult to train the critic network with more high-dimensional actions as inputs in actor-critic algorithms such as TD3 or SAC. With high-dimensional (or more) actions as inputs, the critic becomes more vulnerable to functional approximation error especially at the initial phase of training where it has only seen a narrow data distribution. Because the actor is trained to output the actions that maximize this parameterized critic network, i.e., $\max_{\theta} Q_{\phi}(s, \pi_{\theta}(s))$, the actor can easily exploit this error, which ends up in a vicious cycle that explodes the value function, e.g., target Q-values > 1e9 in our experiments. On the other hand, using a critic-only algorithm that does not take actions as inputs can avoid this issue -- which is why we build our algorithm upon CQN that does not take high-dimensional actions as inputs.
> > >
> > > ---
> > >
> > > **v2-Q4. for the sake of a better understanding on why action sequences help, wouldn't it be better to investigate toy problems that can be analyzed more deeply?**
> > >
> > > **v2-A4.** Thank you for your suggestion. With regard to *why* action sequences are helpful, our hypothesis in line 21-23 was that action sequences can be helpful as they correspond to meaningful action primitives such as going straight -- and learning the long-term effect of taking these primitive actions will be easier than learning the effect of taking single miniscule actions. Our positive results on position-based control and negative results on torque-based control also supports this hypothesis -- action sequences for position-based control can be seen as more semantically meaningful in that each sequence is a line in a joint space (for joint control) or a coordinate space (for end-effector control) compared to a sequence of torques.
> > >
> > > We completely agree with your point that investigation with toy problems can be helpful -- we will work on it by building a toy experimental setup based on our hypothesis and include the analysis in the final version with illustrative visual figures.
> > >
> > > ---
> > >
> > > **v2-Q5. I would suggest replacing "robot" by "simulated robot" in the title**
> > >
> > > **v2-A5.** Thank you for your detailed suggestion. We will update the title and abstract in the final version to avoid confusing the readers.
> > >
> > > ---
> > >
> > > **v2-Q5. I think figure 7a should show a length over 16 and the conclusion should be updated.**
> > >
> > > **v2-A5.** We will update the figure and the writing to reflect this updated experimental results.
> > >
> > > ---
> > >
> > > **v2-Q6. There is a pattern that the return for length 2 is lower than for length 1 and length 4, Any idea why?**
> > >
> > > **v2-A6.** Thank you for having a very detailed look at our responses for other reviewers as well! Here, it is quite difficult to compare CQN-AS2 and CQN-AS4 as they collapsed without learning meaningful behaviors -- a random policy that outputs random actions can achieve a return up to 30 in this task.
> > >
> > > ---

---

> > > > ### Comment · Reviewer_yZtr · 2025-08-03
> > > > **Thank you for the clarification**
> > > >
> > > > Dear authors,
> > > >
> > > > Yes, I found your new response much clearer than the previous one, I'm now more convinced of the merit of the paper and I'm ready to join the other reviewers on the accept side. I hope this discussion will help you further clarify the message of the paper.

---

### Official Review · Reviewer_YKFp · 2025-06-27

**Clarity:** 3
**Significance:** 3
**Originality:** 2
**Rating:** 4
**Confidence:** 3

**Summary:**

The paper introduces Coarse-to-fine Q-Network with Action Sequence (CQN-AS), a value-based RL method that explicitly predicts Q-values over sequences of actions rather than individual actions. The main claim is that modeling the return-to-go over action sequences leads to more accurate value estimation and better performance, especially in sparse-reward settings. This is supported by empirical results showing that CQN-AS often outperforms baseline methods on tasks from BiGym and RLBench.

A key contribution is the formulation of the value function over sequences, which encourages the agent to reason more effectively about temporally extended behaviors. The authors support this claim through comparative results on several benchmarks and provide ablation studies showing that longer action sequences correlate with improved performance. However, while average performance improves, in some tasks gains are modest or inconsistent, which somewhat weakens the universality of the claim. A deeper analysis of when and why CQN-AS works best would help solidify its impact.

**Questions:**

Some tasks show only marginal gains over baselines. Can the authors provide more detailed per-task results or analysis to clarify where CQN-AS is most effective?

The paper mentions using “17 to 60 demonstrations” per task. How is this number chosen? Performance can vary a lot wrt the number of demos used. Can you clarify the choice?

The ablation on sequence lengths shows significant performance improvements with longer action sequences. Is there an upper bound or threshold beyond which increasing the sequence length causes performance to degrade?

Have the authors considered comparisons to sequence models?

**Ethical Concerns:**

["NO or VERY MINOR ethics concerns only"]

**Final Justification:**

I believe my questions have been addressed, I confirm my score, and I have completed the mandatory acknowledgement.

**Limitations:**

The authors are transparent about the method’s limitations on long-horizon tasks, where predicting the return of long action sequences becomes increasingly difficult. This is a meaningful constraint, especially for complex tasks requiring extended planning or multi-stage behaviors. While CQN-AS shows promise on mid-range horizons, its performance may degrade as the sequence length increases due to compounding uncertainty and training instability. This limits its scalability in more complex environments. It would be interesting to see future extensions that incorporate hierarchical sequence modeling or hybrid approaches that decouple high-level planning from low-level execution. Such strategies could help overcome the bottlenecks associated with long sequence modeling and extend the applicability of the method to broader domains.

**Paper Formatting Concerns:**

No concern.

**Quality:**

3

**Strengths And Weaknesses:**

The paper presents a well-motivated problem and a clearly defined method. Experiments cover a diverse set of challenging sparse-reward tasks from BiGym and RLBench, supported by a reasonable set of ablations.

The paper is clearly written and well-structured. The presentation of ideas, contributions, and results is easy to follow. Ablation studies and visualizations help convey the impact of different design decisions effectively.

The idea of leveraging action sequences for value-based RL is timely and relevant, bridging insights from behavior cloning to reinforcement learning. However, in some cases/tasks, it is not clear that the proposed method significantly outperforms the baselines.

The method builds on existing ideas in sequence modeling and action prediction, applying them in a novel way to Q-value estimation. While the high-level motivation is not entirely new, the specific formulation and empirical exploration are original and interesting.
The method draws inspiration from recent trends in behavior cloning and sequence modeling (e.g., decision transformers), but applies these ideas in a novel way within a value-based RL framework—distinct from purely model-free or policy-gradient approaches. To the best of my knowledge, there is no prior value-based method that directly models Q-values over explicit action sequences as done here. This design is original and well-motivated, representing a meaningful contribution to the space of temporal abstraction and sequential reasoning in RL.

Some experimental design choices are not well justified. For example, the statement that "17 to 60 demonstrations" were used leaves ambiguity: using 17 versus 60 demos can have a large impact on performance, particularly in sparse-reward settings. Clarifying this and providing more control over such variables would strengthen the experimental section.

---

> ### Author Rebuttal · Authors · 2025-07-30
>
> Dear Reviewer YKFp,
>
> We really appreciate your time and effort for reviewing our paper. We provide our responses below.
>
> ---
>
> **Q1: Some tasks show only marginal gains over baselines. Can the authors provide more detailed per-task results or analysis to clarify where CQN-AS is most effective?**
>
> **A1.** We find that CQN-AS tends to be more effective on relatively long-horizon tasks compared to baselines. For instance, results on RLBench’s long-horizon tasks like Open Oven or Take Plate Off Colored Dish Rack support this. On the other hand, we find that CQN-AS can be weaker than baselines on very dynamic tasks such as reach target due to the use of temporal ensemble. Finally, we find that CQN-AS does not exhibit stronger performance on tasks like on BiGym’s flip cutlery task, where we hypothesize all methods fail due to challenges in perception. We will include more detailed discussion in the final version. Thank you for your suggestion!
>
> ---
>
> **Q2: Clarification on “17 to 60 demonstrations” per task**
>
> **A2.** As we’ve written in Appendix C, we used different number of demonstrations for different tasks, but we used the same number of demonstrations for all algorithms within each task. This is because not all human demonstrations in BiGym are not successfully replayed in simulation, and therefore we used demonstrations that we were able to replay in simulations, where the available number of demonstrations was different for each task. Specific number of demonstrations used for each task is available in Table 1, Appendix C. We will clarify this in the caption.
>
> ---
>
> **Q3: Is there an upper bound or threshold beyond which increasing the sequence length causes performance to degrade?**
>
> **A3.** We find that using longer action sequences ($K  > 16$) does not improve performance and sometimes degrade performance. We hypothesize this is because training with too long action sequences can be difficult, offsetting the benefit of learning with action sequences. We also note that the best action sequence length can be different for each dataset that has different action frequencies. We will include the relevant results and discussion in the final version.
>
> ---
>
> **Q4: Comparison to sequence models**
>
> **A4.** While using sequence models like Transformer for RL has a potential to achieve better performance on long-horizon tasks (as shown in offline RL papers like [1,2], we find that GRU is faster to train and leads to stronger performance in our considered setups. We hypothesize this is because it is non-trivial to train Transformer models from scratch using unstable training signals from td learning, which leaves a room for interesting future work.
>
> [1] Chebotar, Yevgen, et al. "Q-transformer: Scalable offline reinforcement learning via autoregressive q-functions." Conference on Robot Learning. PMLR, 2023.
>
> [2] Springenberg, Jost Tobias, et al. "Offline actor-critic reinforcement learning scales to large models." arXiv preprint arXiv:2402.05546 (2024).

---

> > ### Comment · Reviewer_YKFp · 2025-08-06
> >
> > Thank you for the detailed rebuttal. The authors have adequately addressed my concerns, and I appreciate the clarifications provided.

---

### Official Review · Reviewer_5gur · 2025-06-30

**Clarity:** 3
**Significance:** 2
**Originality:** 3
**Rating:** 4
**Confidence:** 4

**Summary:**

This paper introduces the Coarse-to-fine Q-Network with Action Sequence (CQN-AS), a novel reinforcement learning (RL) algorithm. The core idea is to enhance value learning in robotics by training a critic network to predict Q-values for sequences of actions rather than individual actions. This approach is motivated by observations that action sequences lead to lower validation loss in return-to-go prediction and that standard actor-critic methods struggle with value overestimation when using action sequences. Experiments show that CQN-AS successfully avoids this overestimation and demonstrates improved performance on various robotic simulation benchmark environments.

**Questions:**

* The author mentioned that CQN-AS may struggle to “achieve meaningful success rates on long-horizon tasks” (4.1). I think the paper would be made stronger if the authors include some potential experiments showing how some of the potential extensions can make CQN-AS perform even better.
* Can you elaborate on on the baselines details of SAC and TD3 you used in obtaining figure 2b). I also don't entirely agree with your claim that action sequencing causes severe value over estimation from by looking at this plot - it looks from the plot that in fact TD3 and SAC may have even more value overestimation compared to the ones using action sequences.
* In appendix table 3, there are mentions of hyperaparameters for C51. Do you use a distributional critic for both CQN and CQN-AS? If so, it would be nice to know how much a distributional critic helps with addressing value estimation. Given a distributional critic has bounded value, this means that you won't be able to see severe value overestimation as those reported in fig 2b.

**Ethical Concerns:**

["NO or VERY MINOR ethics concerns only"]

**Final Justification:**

Thank the authors for the response. The rebuttals address my points. Please consider incoporating some the suggestions into the final version.

**Limitations:**

Yes.

**Paper Formatting Concerns:**

No.

**Quality:**

3

**Strengths And Weaknesses:**

Strengths

* The design of the CQN-AS is well-motivated by the observation that using action sequences lead to lower critic loss.
Extensive evaluation on the performance of CQN-AS.
* The evaluation is thorough.
* Clear ablation of the individual algorithmic choices and their merits in (4.4).

Weaknesses
* Extension to use action sequences based on CQN looks like an incremental extension so I wouldn’t rate this work highly in terms of of novelty.

---

> ### Author Rebuttal · Authors · 2025-07-30
>
> Dear Reviewer 5gur,
>
> We really appreciate your time and effort for reviewing our paper. We provide our responses below.
>
> ---
>
> **Q1: Limited Novelty of CQN-AS.**
>
> **A1.** While the architectural novelty of CQN-AS may be limited, we would like to emphasize our paper has significant contributions: we provide interesting observations from training popular actor-critic algorithms with action sequences, build a new algorithm based on such observations, and provide experimental results and analysis that could be useful for future research.
>
> ---
>
> **Q2: The paper could be stronger with some potential experiments showing how some of the potential extensions can make CQN-AS perform even better.**
>
> **A2.** As we noted in Section 4.1, CQN-AS struggles to achieve meaningful success rate on some of the long-horizon *tasks that require interaction with delicate objects such as cups or cutlery*). We believe this can be resolved by incorporating a better perception stack, e.g., incorporating a stronger vision encoder and experimenting with higher-resolution visual observations. While we tried to run experiments per your suggestion, i.e., increasing the image resolution and incorporating ResNet-18, we found that it requires a GPU with at least 48GB memory and is extremely slow to train. We will leave this direction of incorporating larger vision encoders in an efficient manner as a future direction.
>
> ---
>
> **Q3: Are TD3-AS and SAC-AS really suffering from value overestimation in Figure 2(b)? Can you provide more details of SAC and TD3 baselines in Figure 2(b)?**
>
> **A3.** We would like to clarify that TD3 and SAC with action sequences indeed suffer from severe value overestimation, as shown in the follow table:
>
> | Method | Episode return | Target Q values |
> |---|---|---|
> | TD3-AS1 | 113.97807 | 9.19 |
> | TD3-AS2 | 4.68642 | 6e9 |
> | TD3-AS4 | 27.52748 | 5e9 |
> | SAC-AS1 | 198.16332 | 16.13 |
> | SAC-AS2 | 16.27714 | 3e9 |
> | SAC-AS4 | 22.638 | 1e9 |
>
> Nonetheless, we agree that the figure can be confusing to readers. We will make the figure clearer to understand. For details on SAC and TD3, we used the standard implementation and hyperparameters for training SAC and TD3 for Figure 2(b), for instance, we used the hyperparameters from HumanoidBench official repository for training SAC. For TD3, we used standard deviation of 0.2 for exploration. We will include these details in the final version as well.
>
> ---
>
> **Q4: Did you use a distributional critic for both CQN and CQN-AS? What’s the effect of the distributional critic on RL with action sequence?**
>
> **A4:** We used C51 for both CQN and CQN-AS. To answer your question, we run additional experiments that don't use C51 for CQN-AS on BiGym’s Saucepan To Hob task, and find that both CQN-AS with distributional critic (86 $\pm$ 19\%) and CQN-AS without distributional critic (82.6 $\pm$ 16\%) achieve similar performance. This result shows that using a critic-only algorithm with discrete actions is indeed more crucial in enabling RL with action sequences compared to the use of a distributional critic.

---

### Official Review · Reviewer_H7zC · 2025-07-03

**Clarity:** 3
**Significance:** 3
**Originality:** 2
**Rating:** 5
**Confidence:** 4

**Summary:**

This paper presents Coarse-to-fine Q-Network with Action Sequence (CQN-AS), a novel value-based multi-step RL algorithm. CQN-AS trains a critic network to output Q-values over discretized action sequences in a coarse-to-fine manner. By generating multiple actions in parallel, it is less affected by compounding errors and makes value prediction easier. By completely removing the actor network, it is more robust to function approximation error. Experiments on multiple challenging benchmarks demonstrate that CQN-AS outperforms baseline algorithms in most tasks, particularly in sparse reward scenarios.

**Questions:**

1. Can CQN-AS be extended to the offline RL setting?

2. Would replacing the GRU with a transformer network improve performance on long-horizon tasks?

3. The paper mentions that removing the actor network prevents the actor from overexploiting the critic, making the algorithm more robust to function approximation errors in the critic. However, when performing Q-learning with discretized actions, the algorithm still selects the action that maximizes the next Q-value, which could also correspond to regions with large approximation errors. Why does this not lead to overestimation in such cases?

**Ethical Concerns:**

["NO or VERY MINOR ethics concerns only"]

**Final Justification:**

I am keeping my recommendation for acceptance.

**Limitations:**

Yes.

**Paper Formatting Concerns:**

None.

**Quality:**

3

**Strengths And Weaknesses:**

### Strengths
1. The authors begin by analyzing two empirical observations: (1) the benefit of multistep actions for return-to-go prediction, and (2) the value overestimation in standard actor-critic algorithms when combined with multistep actions. These analyses effectively motivate their proposed method, resulting in a presentation that is logically coherent and clearly motivated.

2. The writing is clear and the structure is well-organized throughout. Figure 2 effectively illustrates the core idea of the algorithm.

3. The authors conduct extensive experiments to validate the effectiveness of the algorithm, along with comprehensive ablation studies that highlight the contribution of each component.

### Weaknesses
I agree with the authors' acknowledgment of the limitations and weaknesses of their method: (1) inferior performance on long-horizon tasks, (2) lack of real-world robot experiments, and (3) suboptimal results under torque control. However, the algorithm is well-motivated and innovative, and the experimental results strongly support the proposed motivation. I believe this work has the potential to inspire many future studies. In my opinion, it is worthy of acceptance and presentation at NeurIPS.

---

> ### Author Rebuttal · Authors · 2025-07-30
>
> Dear Reviewer H7zc,
>
> We really appreciate your time and effort for reviewing our paper. We provide our responses below.
>
> ---
>
> **Q1: CQN-AS for offline RL?**
>
> **A1.** We find positive results in our preliminary experiments. We combined CQN-AS with Cal-QL [1] and trained it on the dataset that consists of 26 successful demonstrations and 10 failed episodes. We find that CQN-AS + Cal-QL achieves 33 ($\pm$ 6.8) \% while CQN + Cal-QL achieves 7 ($\pm$ 14) \%, which shows CQN-AS can be indeed effective for offline setup.
>
> [1] Nakamoto, Mitsuhiko, et al. "Cal-ql: Calibrated offline rl pre-training for efficient online fine-tuning." Advances in Neural Information Processing Systems 36 (2023): 62244-62269.
>
> ---
>
> **Q2: Would GRU -> Transformer improve performance on long-horizon tasks?**
>
> **A2.** This is an interesting question. While using Transformer for RL has a potential to achieve better performance on long-horizon tasks (as shown in offline RL papers like [2,3]), we find that GRU is faster to train and leads to stronger performance in our considered setups. We hypothesize this is because it is non-trivial to train Transformer models from scratch using unstable training signals from temporal difference learning, which leaves a room for interesting future work.
>
> [2] Chebotar, Yevgen, et al. "Q-transformer: Scalable offline reinforcement learning via autoregressive q-functions." Conference on Robot Learning. PMLR, 2023.
>
> [3] Springenberg, Jost Tobias, et al. "Offline actor-critic reinforcement learning scales to large models." arXiv preprint arXiv:2402.05546 (2024).
>
> ---
>
> **Q3: How does Q-learning with discrete actions avoid severe value overestimation?**
>
> **A3.** As you mentioned, Q-learning with discrete actions may still suffer from overestimation due to max operation during training. However, in this paper, we find that a combination of (i) a critic network that takes high-dimensional action sequences as inputs and (ii) the actor that is trained to output actions using gradients from the critic leads to severe value overestimation problems. We hypothesize this is because the critic network tends to have more function approximation errors from high-dimensional actions and the actor can easily exploit this error. On the other hand, critic-only algorithms can avoid this problem, leading to stable training with action sequences.
>
> To further support our point, we conducted additional toy experiments where we introduced redundant no-op actions for training TD3 and CQN agents on DMC Cheetah Run tasks. Specifically, we trained agents with 6 original actions and 294 no-op actions with [-1, 1] action bounds and used an environment wrapper that slices out no-op actions. In this setup, we found that CQN (critic-only algorithm with discrete actions) is indeed robust to the overestimation problem. In particular, we find that TD3 suffers from severe value overestimation with more no-op actions, while CQN with such no-op actions achieves similar performance with the original CQN agent. This clearly shows that a critic-only algorithm is more robust to training with high-dimensional action spaces.
>
> | Additional No-op actions | Episode Return (CQN) | Target Q-Values (CQN) | Episode Return (TD3) | Target Q-Values (TD3) |
> | :--- | :--- | :--- | :--- | :--- |
> | 0 | 236.61 | 18.84 | 140.67 | 16.25 |
> | 54 | 219.50 | 20.27 | 7.18 | -17.45 |
> | 144 | 185.38 | 16.42 | 0.56 | 1.00E+08 |
> | 294 | 202.94 | 18.66 | 0.27 | 4.00E+08 |
>
> Based on this observation, one potential alternative is to use expectile regression for Q-learning as in IQL [4], which can avoid value overestimation problems by avoiding the explicit maximization step in TD-learning. We will leave this as a future work and include relevant discussion in the final version.
>
> [4] Kostrikov, Ilya, Ashvin Nair, and Sergey Levine. "Offline reinforcement learning with implicit q-learning." International Conference on Learning Representations 2022

---

> > ### Comment · Reviewer_H7zC · 2025-08-05
> >
> > Thank you for your reply, and the no-op experiment is interesting.

---

### Note · Authors · 2025-08-16

Dear Reviewers and Area Chair,

We sincerely appreciate your time and effort in reviewing our paper.

We are delighted to find that all reviewers are positive about our paper following the rebuttal. We are particularly encouraged that the reviewers highlighted: (i) the well-motivated idea of using action sequences for value learning based on empirical observations (H7zC, 5gur, YKFp), (ii) the clear and well-organized writing, and (iii) the strong and thorough experimental results (all).

We are grateful for the valuable feedback and we will be sure to incorporate the additional experiments and discussions to further improve the paper’s clarity and significance.

Sincerely,

Authors

---

### Decision · Program_Chairs · 2025-09-17

**Decision:**

Accept (poster)

**Comment:**

The comments from reviewers do note overall issues with the paper.

1. Extension to use action sequences based on CQN looks like an incremental extension, so I wouldn’t rate this work highly in terms of novelty.
2. However, the method's performance is apparent, yet the authors do not use a reliable method to evaluate the performance.
3. Given the lack of real hardware experiments, the authors should be careful about calling this robot learning, and not just reinforcement learning.

Overall, the paper appears to provide a clean step forward in research; however, it would also benefit from additional references to related work on learning multi-step q-functions.

Kahn, Gregory, et al. "Self-supervised deep reinforcement learning with generalized computation graphs for robot navigation." 2018 IEEE international conference on robotics and automation (ICRA). IEEE, 2018.